# Interpreting and Steering State-Space Models via Activation Subspace Bottlenecks

**Vamshi Sunku Mohan** [1]  **Kaustubh Gupta** [2]  **Aneesha Das** [2]  **Chandan Singh** [3]

## Abstract

State-space models (SSMs) have emerged as an efficient strategy for building powerful language models, avoiding the quadratic complexity of computing attention in transformers. Despite their promise, the interpretability and steerability of modern SSMs remain relatively underexplored. We take a major step in this direction by identifying *activation subspace bottlenecks* in the Mamba family of SSM models using tools from mechanistic interpretability. We then introduce a test-time steering intervention that simply multiplies the activations of the identified bottlenecks by a scalar. Across 7 SSMs and 6 diverse benchmarks, this intervention improves performance by an average of 8.27%, without requiring any task-specific tuning. Finally, we validate that the identified bottlenecks are indeed hindering performance by modifying them to yield an architecture we call Stable-Mamba, which achieves long-context performance gains when retrained from scratch.[1]

## 1. Introduction

State-space models (SSMs) have emerged as a promising alternative to transformer-based models (Gu et al., 2021; 2022a), with the Mamba SSM achieving language modeling performance comparable to similar-sized transformer models (Gu & Dao, 2024) while boasting substantial efficiency improvements. This efficiency arises from the fact that SSM next-token predictions are computed recurrently using a fixed-size hidden state, whereas transformer predictions are computed using a key-value cache that grows linearly with the sequence length.

---

[*]Equal contribution [1]AppViewX, Bangalore [2]Independent Researcher [3]Microsoft Research, Redmond. Correspondence to: Vamshi Sunku Mohan <vamshisunku.mohan@appviewx.com>.

*Proceedings of the 43$^{rd}$ International Conference on Machine Learning*, Seoul, South Korea. PMLR 306, 2026. Copyright 2026 by the author(s).

[1]Code is available at github.com/vanivamshi/activation-subspace-bottlenecks.

While the SSM architecture enables efficiency, it comes with some downsides. For example, Mamba often exhibits poor performance for in-context learning and long context retrieval (Qu et al., 2025). Moreover, the underlying causes of these failures can be difficult to pinpoint because the recurrent structure of SSMs lacks the explicit neuron representations found in transformers, which have been the subject of a great deal of interpretability research (Rai et al., 2024; Bills et al., 2023; Bricken et al., 2023; Braun et al., 2025; Bushnaq et al., 2025; Singh et al., 2024). This reduced interpretability positions SSMs as a more opaque architecture, limiting our understanding of them and hindering their deployment and subsequent improvement.

To meet this challenge, we propose a mechanistic investigation of SSMs, largely focused on Mamba. Our approach centers on finding *Activation Subspace Bottlenecks*, by identifying a small set of activation subspaces causally selected from a larger pool critically routing information.

This approach replaces transformer-specific interpretations focused on neurons and attention heads with a steering-focused exploration of recurrent activation subspaces. With this approach, we find that information in Mamba is routed through a small number of dominant activation subspaces, which creates bottlenecks that limit performance (e.g. see Fig. 1a).

Having identified these bottlenecks, we propose a simple, targeted steering intervention to improve performance without requiring fine-tuning by scaling activations at particular points in an SSM. This steering intervention alleviates previously identified bottlenecks and improves performance across standard benchmarks (see Fig. 1b). Surprisingly, this intervention which is identified only using samples from the Pile, improves performance on 7 different SSMs across diverse text benchmarks by an average of 8.27% without requiring any task-specific tuning.

As a secondary evaluation, to validate that the identified bottlenecks are indeed hindering performance, we make minimal modifications to Mamba to yield an architecture we call Stable-Mamba. Stable-Mamba adds only 256 new parameters and has a negligible effect on inference speed and memory consumption. After retraining from scratch,

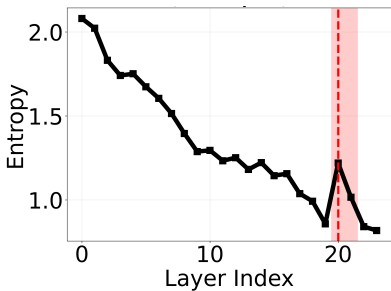

*(a)* Vanilla Mamba
NIAH: 84.00%, QA: 66.7%, Pathfinder: 60.0%

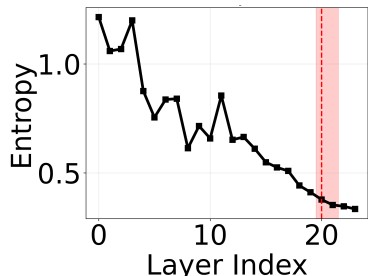

*(b)* Steered Mamba
NIAH: 92.00%, QA: 76.0%, Pathfinder: 74.0%

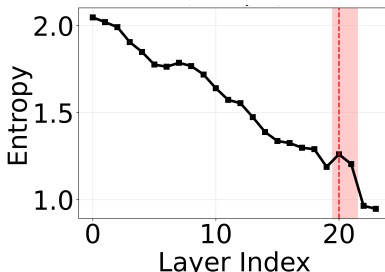

*(c)* Stable-Mamba
NIAH: 90.00%, QA: 82.0%, Pathfinder: 95.0%

*Figure 1.* Entropy across layers measured using Stochastic Parameter Decomposition (Bushnaq et al., 2025). (a) Vanilla Mamba exhibits a sharp entropy spike at Layer 20, indicating a parameter-level routing bottleneck where diverse information is forced through a narrow subset of parameters. After (b) steering and (c) architectural modifications, the spike is removed and entropy becomes smoother, indicating restored information flow. These modifications yield improved performance across three standard benchmarks; NIAH: Needle in a haystack, QA: question-answering and Pathfinder: long-context benchmark.

Stable-Mamba partially avoids the previously identified bottleneck and improves performance across standard benchmarks (Fig. 1c). Taken together, our results show how interpretability methods can be made practically useful by localizing model failures in order to improve them.

**Conflict of Interest Disclosure**   The authors declare no competing financial interests.

## 2. Related Work

**State-Space Models.**   SSMs are linear recurrent architectures with fixed-size hidden states that enable linear-time training and constant-time inference per step, making them a more efficient alternative to transformers for sequence modeling (Gu et al., 2021; 2022a). Recent models like Mamba (Gu & Dao, 2024), Mamba-2 (Dao & Gu, 2024) and others (Zuo et al., 2022; Poli et al., 2023; Wang et al., 2024a; Pióro et al., 2024; Peng et al., 2023) have shown that SSMs can match transformers of similar scale on standard NLP benchmarks, aided by long-range initialization techniques from HiPPO theory (Gu et al., 2022b) and time-dependent parameterization that allows selective attention over inputs. Hybrid transformer-SSM architectures have shown gains in both efficiency and accuracy over pure transformers (Wang et al., 2025; Glorioso et al., 2024), reinforcing the value of SSM-based models, especially as long contexts become increasingly widespread.

**Interpreting SSMs.** Recent work has explored information representation and processing in SSMs with some work investigating universality by uncovering shared circuits across transformers and Mamba (Wang et al., 2024b; Chen et al., 2025b) and others analysing what information is compressed in the latent state of SSMs (Hossain et al., 2025; Jelassi et al., 2024; Wang et al., 2024c). Similarly, a line of mechanistic interpretability work has probed the functionality of differ-

ent model components, e.g. localizing facts (Sharma et al., 2024) or using ablation-based analyzes to examine information flow across layers (Ensign & Garriga-Alonso, 2024; Endy et al., 2025; Jafari et al., 2024). Another work (Paulo et al., 2024) applies different mechanistic interpretability methods, such as the tuned lens (Belrose et al., 2023) and contrastive activation addition (Rimsky et al., 2024), to understand and elicit different behaviors from Mamba.

Despite these studies, interpretability research has led to few practical improvements in SSMs, despite a broad set of effective steering methods introduced for transformers (Subramani et al., 2022; Zhang et al., 2023; Zhan et al., 2025; Mallen et al., 2023; O'Brien et al., 2024). We make connections between SSMs and many popular interpretation techniques, including neuron-level interpretation (Gurnee et al., 2024; Song et al., 2024; Dai et al., 2022; Singh et al., 2023b), sparse autoencoders (Shu et al., 2025) and parameter-level decomposition (Bushnaq et al., 2025; Braun et al., 2025).

**Comparing SSMs with Prior Architectures.**   Another set of related works makes comparisons between SSMs and existing architectures. Han et al. (2024) interpret Mamba's linear attention as analogous to the input and forget gates in LSTMs (Hochreiter & Schmidhuber, 1997), suggesting a gating-based view of recurrent dynamics. More recently, several studies have identified attention-like behavior in SSM hidden states (Ali et al., 2025b; Zimerman et al., 2024; Ali et al., 2025a), showing that implicit sequence-dependent interactions can be represented as decoder-style attention masks.

## 3. Methods

In this section, we provide an overview of *Activation Subspace Bottlenecks*, which consists of a set of tools to interpret the different subspaces present in SSM internal repre-

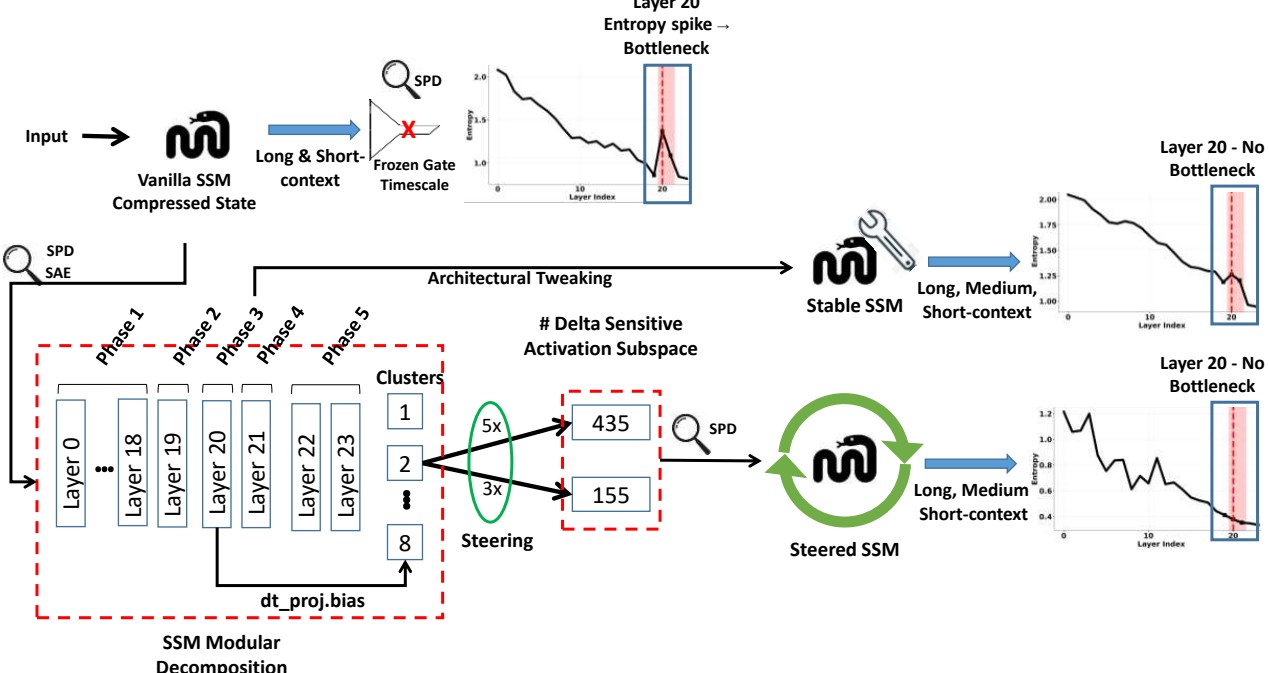

*Figure 2. Workflow for identifying Activation Subspace Bottlenecks* and using them to conduct post-hoc steering or to make architectural modifications to Mamba.

sentations (Section 3.1). Next, we show how the shortcomings identified by these bottlenecks can be mitigated via steering (Section 3.2) or through architectural modifications (Section 3.3). Fig. 2 shows an overview of this entire workflow and we provide a comprehensive report on interpreting Mamba's internals in Appendix A-Appendix C.

## 3.1. Identifying Activation Subspace Bottlenecks

### 3.1.1. QUALITATIVELY EXPLAINING ACTIVATION SUBSPACE BOTTLENECKS

To motivate identifying Activation Subspace Bottlenecks, we begin by qualitatively characterizing how computation in Mamba is organized. For this purpose, we train Sparse Autoencoders (SAEs), which are learned neural networks that reconstruct hidden activations using sparse latent representations, yielding latent features that define activation-level feature discovery. SAE is used as compressive bottleneck to obtain a compact, low-dimensional summary of activations capturing dominant variation (Goodfellow et al., 2016). Rather than learning a large monosemantic dictionary, it acts as a low-rank projection. Dictionary learning (Mairal et al., 2009) is applied on SAE latent representations to recover interpretable components and quantify feature utilization using sparsity, entropy, coefficient of variation and KL divergence.

### 3.1.2. QUANTITATIVELY IDENTIFYING ACTIVATION SUBSPACE BOTTLENECKS

**Defining activation subspaces.** We now seek to identify Activation Subspace Bottlenecks that can be used for steering SSMs. Since Mamba does not contain explicit attention heads that can be interpreted and steered directly (Bibal et al., 2022; Zhang et al., 2023), we instead interpret activation subspaces by projecting hidden states onto attention-weighted vectors derived from implicit attention matrices.

Specifically, for each layer, we construct an attention-style matrix $A \in \mathbb{R}^{H \times T \times T}$ from the Mamba *mixer.ssm* layer using the attention-mapping procedure of Ali et al. (2025b). For each head (h), the causal token-to-token interaction is given by $\alpha_{t,s}^{(h)} = \sum_m Q_t^{(m,h)} H_{t,s}^{(m,h)} K_s^{(m,h)}, \quad s \leq t$, which measures the influence of source token (s) on target token (t) under the SSM dynamics.

where, $T$ = sequence length, $H$ = number of attention heads in the attention-mapping approximation, $m$ = internal state dimension, $Q^{(m,h)}, K^{(m,h)}$ = query and key derived from the SSM parameters.

We then aggregate across heads to obtain a token-to-token influence matrix:

$$A = \frac{1}{H} \sum_{h=1}^{H} \alpha^{(h)}, \quad A \in \mathbb{R}^{T \times T}. \tag{1}$$

From this matrix we derive a token-level importance vector which measures how strongly each token position contributes to the recurrent state updates:

$$w = \frac{1}{T} \sum_{j=1}^{T} A_{:,j}, \quad w \in \mathbb{R}^T \tag{2}$$

We shift and scale the vector by its minimum and maximum, to obtain a normalized vector $\tilde{w}$ with values in the range $[0, 1]$.

$$\text{Activation subspace} = \sum_{t=1}^{T} \tilde{w}_t \cdot h_t, \tag{3}$$

where $h_t \in R^d$ is the SSM hidden state at time step $t$ for a particular layer.

Activation subspaces are weighted sums of hidden states producing a single representation-space vector summarizing token-level contributions. Further, *feature subspaces* are low-dimensional SAE-learned directions capturing structured components of the activation space.

We next apply a 2-stage procedure described below to pinpoint functional bottlenecks within SSMs using these activation subspaces.

**Parameter-level decomposition.** We first make use of Stochastic Parameter Decomposition (SPD) which decomposes model parameters into a sum of sparsely used vectors that satisfy various empirically useful properties (Bushnaq et al., 2025). SPD provides parameter-level metrics to identify activation subspace bottlenecks, such as the activation mean and variance, sparsity (fraction of near-zero activations), entropy (measuring information diversity of representations, as shown in Fig. 1), effective rank (dimensional utilization of the representation space), coefficient of variation (relative variability of parameter sensitivity), gradient sensitivity (magnitude of optimization response) and post-ablation KL divergence (information loss upon ablation).

**Delta-Sensitive subspaces.** To isolate recurrent components that control state updates, we introduce Delta-Sensitive subspaces. In Mamba, $\Delta$ parameterises the discretised time-step of the state-space model, determining how strongly new inputs update the recurrent hidden state (Gu & Dao, 2024). Subspaces whose activations vary strongly with these $\Delta$-mediated updates reflect input-dependent recurrent dynamics and are therefore used as steering targets. We identify Delta-Sensitive subspaces by recording the values for activation subspaces (Eq. (3)) during forward pass from the SSM recurrent block (*mixer.ssm*) across a large text corpus. We then classify the subspaces with the highest variance as delta-sensitive.

## 3.2. Post-hoc Steering of Activation Subspace Bottlenecks

To overcome the structural limitations identified in Section 3.1, we perform post-hoc steering. Our post-hoc steering simply amplifies the values of particular activations at inference time by simply multiplying them by a scalar steering factor. It requires 3 choices: which layer to steer within, which subspaces to steer within that layer and the scalar steering factor:

1. To identify which layer to steer, we compute SPD statistics for each layer and monitor for entropy spikes.
2. To identify which subspaces to steer within a layer, we select Delta-sensitive subspaces by extracting activations from *mixer.ssm* and selecting subspaces exhibiting high variance across inputs in *The Pile* (Gao et al., 2020). We then validate their importance by measuring the effect of their removal on the model's top-1 next-token prediction accuracy. We retain subspaces whose ablation substantially degrades performance on the tuning set when these subspaces are zeroed out. This identifies 668 Delta-sensitive subspaces out of 768 in Mamba.
3. To determine the steering factor, we perform a grid search over values [0.1, 100] and select those that maximize performance on *The Pile*. We find that a steering factor of 5 yields the best performance, followed by a factor of 2, while other values show performance degradation. Delta-sensitive subspaces with performance drop >2% and performance drop $\in (-2\%, 2\%]$ selected via ablation are amplified $5\times$ and $2\times$ respectively.

When we steer other SSM models (see Section 4.1), we apply the same interpretability pipeline to identify bottleneck layers analogous to those identified in Mamba. Post-hoc steering directly targets these bottlenecks without modifying the architecture. Its effectiveness (Fig. 1b) shows that the bottleneck is real and performance-limiting. Delta-sensitive subspaces within the bottleneck layers are then selected using the same variance and ablation criteria and validated through steering sweeps over amplification factors ensuring task-agnostic steering.

## 3.3. Stable-Mamba: Architectural Modifications to Avoid Activation Subspace Bottlenecks

To address long-context limitations identified by the interpretability pipeline in Section 3.1 that remain unresolved through post-hoc steering, we introduce *Stable-Mamba*, a holistic modification of Mamba guided by interpretability insights. While post-hoc steering provides the primary causal evidence for the bottleneck, Stable-Mamba serves as an architecture-level validation of the same failure modes. Rather than targeting the bottleneck in isolation, Stable-Mamba incorporates jointly applied architectural changes addressing the identified failure modes, including the Layer

20 compression bottleneck, extreme sparsity and unstable gradient dynamics, through the following modifications:

- *Multi-timescale State Dynamics.* We introduce parallel state updates at distinct temporal resolutions to enable simultaneous short, medium and long-range processing.
- *Ensembled Output.* A weighted ensemble replaces the linear output projection with adaptive temporal resolution.
- *Sparse Global Context Injection.* Sparse attention injects global summaries into local state representations, stabilising information flow.
- *Learned Gating.* Ensembled gates enable selective activation of task-relevant features, reducing extreme sparsity.
- *Adaptive Compression Strength.* Compression is dynamically modulated for phase-specific capacity control.
- *Gradient Scaling.* Stabilizes deep SSM chains.
- *Scaled Residual Connections.* Decoupled residual scaling improves gradient flow and feature retention.

They are motivated by the more thorough mechanistic analysis of Mamba in Appendix A-Appendix C. The changes result in 256 more parameters and have a negligible effect on the time or memory needed for inference (Table E4) due to efficient parallelization of the added computations despite increased per-token computation. Stable-Mamba is trained from scratch on *The Pile*. See detailed architecture and training details in Appendix E.

# 4. Results

## 4.1. Experimental Setup

**Identifying bottlenecks setup.** We conduct our interpretability analyzes (Section 4.3) using the *Wikitext-2-v1* training split with 33k samples (and perform validation on the test split of 3.9k samples where relevant) using Mamba-130M (Gu & Dao, 2024). When comparing to a transformer, we compare against GPT-2 Small with 124M parameters, trained on *Wikitext-2-v1*.

When training SAEs, we use a symmetric encoder-decoder architecture, where the encoder is a 2-layer fully connected network that maps from an embedding dimension of 768 to a hidden dimension of 460, through a ReLU nonlinearity and then to a sparse embedding dimension of 230. We then process the SAE latents using dictionary learning with a dictionary size of 512, a sparsity penalty ($\alpha$) of 1.0 and 500 training iterations.

**Steering setup.** Steering hyperparameters are tuned using 50k samples from *The Pile*. Steering performance is evaluated on the test set of 6 diverse benchmarks: *TriviaQA* (Joshi et al., 2017), *SQuAD* (Rajpurkar et al., 2016), MuSiQue (Trivedi et al., 2022), IFEval (Zhou et al., 2023), RULER (Hsieh et al., 2024) and DROP (Dua et al., 2019). We further evaluate steering performance for 5 SSM models besides

Mamba-130M: Steered Mamba, Stable-Mamba, Mamba-2 (Dao & Gu, 2024), DenseMamba (He et al., 2024), Hyena (Poli et al., 2023) and MiniPLM-Mamba-130M (Gu et al., 2024). Steering hyperparameters are tuned once for each model (using *The Pile*) and thus require no task-specific tuning.

**Stable-Mamba setup.** We train Stable-Mamba on *The Pile* from scratch and evaluate it alongside the other models on three long-context benchmarks: RULER (Hsieh et al., 2024), Long Range Arena (Tay et al., 2020) and LongContext V2 (Bai et al., 2025). Besides comparing against the SSMs above, we also make compare with GPT-2 Small with 124M parameters (Radford et al., 2019).

## 4.2. Evaluation Metrics

We define evaluation metrics to analyze model behavior and performance.

- **Entropy** - Measures uncertainty in the SPD-based normalized attribution distribution at each layer, indicating how concentrated the distribution is
- **Sensitivity** - Measures the change in model output under controlled input perturbations
- **Correlation** - Pearson correlation (Berman, 2016) between layer-wise signals and interpretable properties.
- **Universality** - Defined as $U = \mu/(1 + \sigma^2)$, capturing the consistency of a metric across inputs
- **Causal Effect** - Quantified using KL divergence between model outputs with and without layer ablation
- **Delta-sensitivity** - Quantifies activation variance induced by input perturbations ($\Delta$)

## 4.3. Activation Subspace Bottlenecks results

### 4.3.1. PRELIMINARIES: QUALITATIVELY CHARACTERIZING ACTIVATION SUBSPACE BOTTLENECKS

To motivate identifying and using Activation Subspace Bottlenecks, we begin by qualitatively characterizing how computation in Mamba is organized.

**Probing activation subspaces for factual knowledge.** To assess whether subspaces compactly store factual knowledge, we perturb delta-sensitive subspaces and measure the resulting change in perplexity (Table 1) across a standard set of factual relations used in prior probing work (Dai et al., 2022). Following Dai et al. (2022), perplexity is computed on cloze-style factual prompts (i.e., predicting masked entities over the full vocabulary) and is therefore not directly comparable to standard language modeling perplexity.

Perturbations in Mamba lead to a 394.1% increase in perplexity, whereas Transformer perplexity remains comparatively stable. This stark contrast indicates that factual predic-

*Table 1. Perplexity in the Transformer and Mamba before and after ablating different knowledge-related subspaces.* Cloze-style prompts on masked answer tokens are used (Dai et al., 2022), yielding higher values than standard next-token perplexity (often >30). Mamba shows substantially larger values, suggesting knowledge is encoded in relatively few overlapping subspaces.

| Relation | PPL (Transformer) | | PPL (Mamba) | |
|---|---|---|---|---|
| | Before | After | Before | After |
| P264 (record label) | 189.60 | 188.07 | 77.30 | 176.15 |
| P449 (original network) | 77.06 | 74.41 | 44.60 | 220.37 |
| P413 (position played on team) | 142.04 | 170.06 | 29.68 | 63.08 |
| P463 (member of) | 15.54 | 9.76 | 9.46 | 24.95 |
| P530 (diplomatic relation) | 31.29 | 24.81 | 15.39 | 52.41 |
| P30 (continent) | 66.07 | 36.49 | 35.22 | 51.46 |
| P36 (capital) | 121.26 | 79.05 | 40.21 | 231.09 |
| P495 (country of origin) | 138.64 | 168.64 | 125.77 | 286.78 |
| P279 (subclass of) | 110.27 | 103.79 | 71.01 | 139.47 |

*Table 2. Correlation analysis of information types in SAE-derived features with encoded information strength.*

| Information Type | Result | Analysis |
|---|---|---|
| Activation Strength | 0.87 | High correlation - Model explicitly tracks activation strength |
| Token Index | 0.65 | Moderate correlation - Positional information distributed across subspaces 128, 256, 384, 512 and 640 at regular intervals of 128 |
| Sparsity | 0.45 | Lower sparsity - Information is encoded nonlinearly and distributed across dimensions |

tion in Mamba is strongly influenced by recurrent dynamics. This observation motivates us to localize the specific layers and activation subspaces responsible for this behavior, since targeted modifications to these components could induce large behavioral shifts.

**Probing activation subspaces for sparse, intervenable feature representations.** Motivated by the finding that factual prediction in Mamba is strongly shaped by recurrent dynamics, we apply SAEs to extract latent features such as, activation strength, token position and sparsity, from hidden activations (Table 2), highlighting which aspects of the representation are most influential and identifying interpretable subspaces for intervention. While SAEs identify important features, they do not capture their utilization within the representation. Dictionary learning addresses this by decomposing SAE latents into interpretable components and quantifying feature usage and efficiency (Cho et al., 2025). Together with SPD (Section 3.1) localizing bottlenecks (e.g., Layer 20 in Fig. 1), these components provide a complementary view of utilization and failure modes. Activation strength has the largest effect (0.87), making it the primary feature for steering delta-sensitive subspaces to improve performance. Dictionary learning further supports this decomposition, showing that feature usage, reconstruction error and sparsity occupy distinct latent dimensions (Table 3), revealing functional specialization and providing a mechanistic handle for targeted interventions.

Steering partially redistributes sensitivity, reducing KL divergence but not fully eliminating the bottleneck. In contrast, Stable-Mamba exhibits more uniform sensitivity and substantially reduced KL at Layer 20, suggesting effective mitigation of this compression point.

### 4.3.2. QUANTITATIVELY IDENTIFYING ACTIVATION SUBSPACE BOTTLENECKS

Having motivated the search for useful subspaces in SSMs, we now quantitatively search for activation bottlenecks that may be useful for steering. To do so, we first employ SPD, which provide layer-wise, parameter-level metrics (see description in Section 3.1.2).

**Finding bottlenecks at the level of entire layers.** We begin by identifying high-level functional roles and bottlenecks at the layer level. We compute SPD entropy across layers and find that it has a large peak in Layer 20 of Mamba (Fig. 1a), suggesting the presence of a parameter-level routing bottleneck, where diverse information is forced through a narrow subset of parameters.

Quantitative SPD statistics (Table 4) provide a nuanced view of layer dynamics. Layers 0–18 show high entropy (1.04), low variance (2.0) and moderate rank (5.85), indicating diverse information with stable activations. Hence, it corresponds to Mamba's sparse feature extraction and gradual accumulation. Layer 19 has slightly lower entropy but higher variance and rank, indicating emerging specialization and variable activation subspace responses, hence mapped to preliminary reorganization before the main bottleneck. Layer 20 exhibits the highest entropy, variance and rank, indicating concentrated routing through a narrow parameter subset, indicating bottleneck. Layer 21 shows lower entropy but higher variance and rank, indicating selective feature amplification, hence mapped to transitional processing post-bottleneck. Further, Layers 22 and 23 have lowest entropy and highest variance, indicating focused but variable activations, hence mapped to progressive feature transformation for output projection. These metrics quantify layer-level bottlenecks without over-interpreting specific functions.

We then further zoom in on Layers 19, 20 and 21. SPD results in Table 5 reveal a pronounced bottleneck at Layer 20 in Vanilla Mamba, evidenced by low gradient sensitivity and extremely high post-ablation KL divergence. Steering partially redistributes sensitivity, reducing KL divergence but not fully eliminating the bottleneck. In contrast, Stable-Mamba exhibits more uniform sensitivity and substantially reduced KL at Layer 20, suggesting effective mitigation of this compression point. A finer-grained SPD decomposition (Appendix B.8) attributes much of this bottleneck to the $dt\_proj.bias$ ($b_{dt}$) parameter governing the SSM time constant $\Delta t = \tau_\Delta(b_{dt} + s_\Delta(x))$ (Gu & Dao, 2024). This

*Table 3. SAE dictionary statistics for Vanilla, Steered and Stable Mamba. Metrics report dictionary size, reconstruction error, sparsity, feature usage, and active features Steered and Stable-Mamba substantially reduce sparsity while lowering reconstruction error, indicating denser and more distributed representations.*

| Metric | Vanilla Mamba | Steered Mamba | Stable-Mamba |
|---|---|---|---|
| Reconstruction error | 9404.5 | 238.2 | 267.6 |
| Sparsity (%) | 98.12 | 62.05 | 58.37 |
| Active features (%) (Activation $> 0.1$) | 18.75 | 60.00 | 68.95 |
| Perplexity (%) | 36.77 | 28.83 | 27.50 |

*Table 4. Layer-level functional roles based on SPD statistics.*

| Phase | Layers | Entropy | Variance ($\sigma$) | Rank |
|---|---|---|---|---|
| 1 | 0-18 | 1.04 | 2.0 | 5.85 |
| 2 | 19 | 1.02 | 2.30 | 6.25 |
| 3 | 20 | 1.19 | 2.63 | 7.59 |
| 4 | 21 | 0.92 | 3.59 | 6.49 |
| 5 | 22-23 | 0.70 | 11.66 | 6.43 |

parameter exhibits strong selective suppression, governs long-range temporal dynamics and concentrates information into a small number of recurrent states, contributing to the observed bottleneck.

**Finding steerable bottlenecks within the identified layer.** To further refine Layer 20 into a set of steerable parameters, we identify activations influencing SSM dynamics. Specifically, we screen subspaces for delta sensitivity (see Section 3.1), identifying the 668 most delta-sensitive subspaces after thresholding sensitivity at 0.01.

We select subsets of these 668 delta-sensitive subspaces to steer by measuring next-token top-1 prediction accuracy on *The Pile* when each subspace is individually ablated by setting their activation values to 0. Intuitively, a more important subspace should yield a larger drop in performance. We find a range of performance drops for different subspaces, with 435 of the 668 delta-sensitive subspaces (65.11%) incurring a drop of atleast 2% and another 155 of the subspaces incurring little to no drop (between -2% and +2%); see details in Appendix D. After performing a cross-validation sweep over different steering factors, we select to amplify the top-435 subspaces with a steering factor of 5 and the following 155 subspaces with a steering factor of 2.

### 4.4. Post-Hoc Steering Performance

Fig. 3 shows that our steering intervention consistently improves performance for Mamba across 6 diverse benchmarks, including long-context evaluation on RULER, with an average improvement of 23.32% and the biggest improvement (17.5%) coming on *IFEval*, an instruction-following dataset. Interestingly, the same tuned hyperparameters generalized across all tested benchmarks, suggesting that steering improvements are not task-specific.

Our steering approach generalizes across 7 SSM models, yielding an average improvement of 8.27% across benchmarks. This includes several saturated or no-improvement cases (e.g., DenseMamba on SQuAD and Mamba-2 on TriviaQA as seen in Fig. 3) where further gains are not possible. Excluding these cases, performance improves by 23.0%.

Steering also mitigates the entropy bottleneck identified by SPD in Vanilla Mamba (see Fig. 1b) and yields less sparse features, measured via SAE dictionary statistics (Table 3). Long-context improvements are further validated across RULER, Long Range Arena and LongBench v2 in Table 7; see extended results in Appendix D.

**Ablations on steering performance.** Our steering pipeline uses interpretable metrics to select both (i) a specific layer and (ii) activation subspaces within that layer as the steering target. To validate these choices, we compare steering interventions applied at different layers and across different subspaces using the same steering strength. Table 6 (top) shows that steering Delta-sensitive subspaces at Layer 20 preserves baseline IFEval accuracy, whereas steering Delta-sensitive subspaces in other layers leads to noticeable performance degradation. Next, Table 6 (bottom) shows that when steering is applied to random and high variance subspaces within Layer 20, Delta-sensitive subspaces yields the strongest performance, while steering random or high-variance subspaces degrades accuracy. Layer and subspace-level ablation indicate that high-variance, low-variance, random and uniform activation interventions degrade performance, whereas only Delta-sensitive subspaces preserve accuracy.

### 4.5. Stable-Mamba

**Stable-Mamba benchmark performance.** Table 7 shows benchmark performance for *Stable-Mamba* against the Vanilla Mamba model, GPT-2 Small and other SSM models on the three long-context benchmarks. All reported columns follow the original benchmark definitions for RULER, Long Range Arena, and LongBench v2. Stable-Mamba yields consistent gains over the base model across the three long-context benchmarks, with the largest gains coming from the QA task within RULER (+15.3 pp) and the PathFinder task within Long Range Arena (+35 pp). Stable-Mamba also out-

*Table 5. Parameter sensitivity and information compression for Layers 19-21 using SPD.* A higher coefficient of variation indicates gradient stability, a high gradient norm indicates high optimization sensitivity and KL divergence quantifies information loss. Vanilla Mamba shows a Layer 20 bottleneck (low gradient, high KL). Steering partially redistributes sensitivity but retains elevated KL, while Stable-Mamba exhibits more uniform sensitivity, higher gradients, and reduced KL, indicating bottleneck mitigation.

| Metric | Mamba | | | Steered Mamba | | | Stable-Mamba | | |
|---|---|---|---|---|---|---|---|---|---|
| | L19 | L20 | L21 | L19 | L20 | L21 | L19 | L20 | L21 |
| SPD Coefficient of Variation | 0.346 | 0.067 | 0.526 | 0.031 | 0.165 | 0.019 | 0.0017 | 0.0098 | 0.0018 |
| Gradient Sensitivity Norm | 0.811 | 0.072 | 0.790 | 0.076 | 0.076 | 0.075 | 0.153 | 0.101 | 0.096 |
| Post-Ablation KL Divergence | 1.0 | 813.0 | 1.0 | 1.0 | 65.04 | 1.0 | 1.8 | 51.87 | 1.0 |

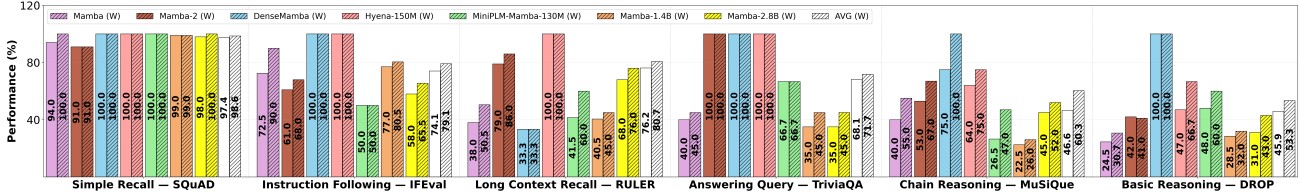

*Figure 3. Performance comparison of SSM models with and without transferred steering parameters.* Models are trained on *The Pile* and evaluated on test sets of task-specific benchmarks: *SQuAD* (recall), *IFEval* (instruction following), *RULER* (long context), *MuSiQue* (multi-hop reasoning), *DROP* (basic reasoning) and *TriviaQA* (QA), representing query types where SSMs show strong performance (Guan et al., 2025; Wang et al., 2025). Solid and striped bars denote performance without and with steering, respectively.

*Table 6. Ablations of steering target selection by layer and by subspace.* Values show next-token prediction accuracy for Mamba on IFEval. We find that both the selection of layer (top section) and filtering by delta-sensitive subspaces (bottom section) is important for achieving high accuracy when steering. Baseline accuracy for Vanilla Mamba is 72.50%.

| Steering Target | Accuracy (%) |
|---|---|
| Layer 18 | 65.0 |
| Layer 19 | 69.0 |
| **Layer 20** | **72.5** |
| Layer 21 | 59.0 |
| Layer 22 | 61.0 |
| **Delta-sensitive Subspaces** | **72.5** |
| Random Subspaces | 67.0 |
| High-Variance Subspaces | 33.5 |
| Low-Variance Subspaces | 46.5 |
| Uniform steering across subspaces | 61.5 |

performs the other compared models. After Stable-Mamba, the best performing model on average is Steered Mamba, which often only lags slightly behind despite not requiring any explicit training.

**Interpreting Stable-Mamba.** Moving beyond prediction performance, we interpret Stable-Mamba and find that it mitigates some of the Activation Subspace Bottlenecks we had previously observed in Vanilla Mamba. To start, it mitigates the bottleneck present in the SPD entropy (Fig. 1c), reducing entropy by 22% in the layer 20 bottleneck. Similar to Steered Mamba it reduces sparsity and increases feature

utilization, as measured through SAE dictionary statistics (Table 3). This suggests better allocation of the model's capacity at inference time. Finally, Stable-Mamba yields general decreases in the SPD coefficient of variation (see Table 5), suggesting that its training dynamics are more stable. A more detailed analysis is provided Appendix E.

## 5. Discussion

Our work provides a mechanistic interpretation of Mamba through the lens of Activation Subspace Bottlenecks. These bottlenecks expose structural limitations in SSM's design which are addressable by selectively steering important activation subspaces. These results showcase one way that interpretability can serve as a practical tool for targeted intervention.

**Limitations.** Our work focuses fairly narrowly on the Mamba model (although we do generalize our steering method to other SSMs) and it remains unclear whether the same pipeline could generalize to other architectures, particularly at very large parameter sizes. We additionally focus on interpreting a few metrics to stabilize recurrent dynamics within Mamba, but do not rule out that alternative metrics may yield stronger results. Stable-Mamba incorporates multiple changes (detailed in Appendix C), but each part is not ablated independently due to compute constraints. Finally, the set of tasks we evaluate on is fairly standard, but it is unclear whether our methods generalize to other types of tasks, e.g. large-scale reasoning.

*Table 7. Performance comparison across long-context benchmarks.* Column names follow the original benchmark definitions. RULER uses a synthetic context length of 1000 tokens, LongBench v2 truncates inputs to 256 tokens and LRA uses a fixed sequence length of 100 tokens. RULER and LRA report task-level performance, while LongBench v2 reports benchmark-defined metrics aggregated across tasks. Stable-Mamba variants achieve the strongest performance, with Steered Mamba consistently outperforming their vanilla counterparts.

| | RULER | | | | Long Range Arena | | | | | LongBench v2 | | | | | |
|---|---|---|---|---|---|---|---|---|---|---|---|---|---|---|---|
| Model | NIAH (%) | Aggrega-tion (%) | QA (%) | **AVG (%)** | ListOps (%) | Retrie-val (%) | Image (%) | Pathfi-nder (%) | **AVG (%)** | Accur-acy (%) | Confide-nce (%) | Calibra-tion (%) | Faithf-ulness (%) | Robust-ness (%) | **AVG (%)** |
| Mamba-130M | 84.00 | 67.00 | 66.70 | 72.57 | 62.00 | 61.00 | 25.00 | 60.00 | 52.00 | 100.00 | 21.99 | 96.86 | 100.00 | 100.00 | 83.77 |
| Mamba-1.4B | 87.71 | 71.00 | 94.00 | 84.24 | 65.00 | 68.00 | 26.50 | 65.00 | 56.13 | 100.00 | 24.22 | 88.95 | 100.00 | 100.00 | 82.63 |
| Mamba-2.8B | 90.43 | 70.00 | 95.00 | 85.14 | 74.00 | 69.00 | 49.50 | 70.00 | 65.63 | 100.00 | 26.55 | 95.63 | 100.00 | 100.00 | 84.44 |
| Steered Mamba-130M | 92.00 | 67.00 | 76.00 | 78.33 | 73.00 | 61.00 | 35.50 | 74.00 | 60.88 | 100.00 | 36.61 | 99.73 | 100.00 | 100.00 | 87.27 |
| Steered Mamba-1.4B | 91.00 | 72.00 | 95.50 | 86.17 | 78.00 | 68.10 | 38.20 | 70.00 | 63.58 | 100.00 | 38.33 | 91.00 | 100.00 | 100.00 | 85.87 |
| Steered Mamba-2.8B | 94.50 | 70.00 | 96.00 | 86.83 | 81.00 | 68.50 | 50.50 | 70.00 | 67.50 | 100.00 | 40.42 | 95.02 | 100.00 | 100.00 | 87.09 |
| Stable-Mamba-130M | 90.00 | 73.00 | 82.00 | 81.67 | 80.00 | 60.00 | 37.00 | 95.00 | 68.00 | 100.00 | 38.35 | 98.92 | 100.00 | 100.00 | 87.45 |
| Stable-Mamba-1.4B | 91.55 | 100.00 | 100.00 | 97.18 | 83.00 | 70.00 | 42.60 | 77.00 | 68.15 | 100.00 | 42.14 | 91.17 | 100.00 | 100.00 | 86.66 |
| Stable-Mamba-2.8B | 93.75 | 100.00 | 99.00 | **97.58** | 85.50 | 70.00 | 55.50 | 100.00 | **77.75** | 100.00 | 45.32 | 96.11 | 100.00 | 100.00 | **88.29** |
| GPT-2 | 94.00 | 59.00 | 58.00 | 70.33 | 64.00 | 20.00 | 66.20 | 100.00 | 62.55 | 95.00 | 12.18 | 81.68 | 95.00 | 95.00 | 75.37 |
| Hyena | 44.14 | 13.50 | 41.00 | 32.88 | 74.00 | 20.00 | 26.00 | 100.00 | 55.00 | 100.00 | 14.85 | 89.78 | 100.00 | 100.00 | 80.93 |
| DenseMamba | 62.00 | 45.00 | 78.00 | 61.67 | 50.00 | 40.60 | 43.00 | 95.00 | 57.15 | 80.00 | 16.63 | 82.41 | 80.00 | 100.00 | 71.81 |
| Mamba-2 | 70.43 | 40.00 | 89.00 | 66.48 | 67.00 | 38.00 | 38.00 | 1.00 | 36.00 | 100.00 | 16.71 | 92.82 | 100.00 | 100.00 | 81.91 |
| MiniPLM-Mamba-130M | 46.00 | 36.00 | 77.00 | 53.00 | 48.00 | 5.00 | 74.50 | 4.00 | 32.88 | 100.00 | 13.55 | 87.44 | 100.00 | 100.00 | 80.20 |

**Future work.** A natural extension of this work is to study SSMs in domains outside of text where they perform strongly, such as computer vision (Huang et al., 2025; Hu et al., 2024; Ma et al., 2024; Xing et al., 2024) and time-series modeling (Yu et al., 2025). Our framework could reveal whether similar Activation Subspace Bottlenecks govern performance across modalities.

Another direction could connect our findings to the growing body of mechanistic interpretability research. For example, it could improve SSMs by reverse-engineering any of the findings in our more thorough mechanistic investigations in Appendix A-Appendix C, building in insights from mechanistic components such as induction heads (Olsson et al., 2022; Kim et al.), n-gram heads and representations (Akyürek et al., 2024; Singh et al., 2023a), concept-based models (Sun et al., 2024; Feng et al., 2024) and more (Todd et al., 2025; Singh et al., 2026; Feng et al., 2026). Alternatively, as new interpretation metrics and automated pipelines are proposed, the search for more compact and steerable Activation Subspace Bottlenecks could be improved, as has been the case for areas such as automated circuit finding (Hsu et al., 2024), embedding interpretation (Benara et al., 2024), or SAE interpretation (Arad et al., 2025).

## Impact Statement

This work improves transparency and performance of SSMs through mechanistic interpretability, identifying structural bottlenecks and demonstrating that both post-hoc steering and minor architectural modifications can mitigate key limitations without complete remodeling and retraining. These interventions enable long-context reasoning and efficient information utilisation at reduced computational cost. However, the ability to selectively manipulate internal representations necessitates further evaluation to avoid unintended amplification of fraudulent and biased behaviors. This work contributes tools for safer, more efficient and more interpretable deployment of large sequence models.

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

# A. Extended Attention Mapping Results

In this section, we present extended results for additional attention mapping techniques beyond the Delta-sensitive subspaces described in Section 4.3.

## A.1. Universality subspaces

Universality (Gurnee et al., 2024) defined in Fig. A1 shows that Mamba has a few subspaces with very high delta variance indicating strong task-specific sensitivity and centralised computation, while the rest taper off more gradually. In contrast, Transformer subspaces have lower overall variance, suggesting more stable and uniform behavior across tasks. Further, to analyze how these are overlap and interact with eachother, we project hidden-state embeddings into a 2D PCA as shown in Fig. A2. This analysis provides a geometric view of representations are distributed and reused, enabling comparison between Mamba and Transformer in terms of subspace sharing, task sensitivity and representational stability.

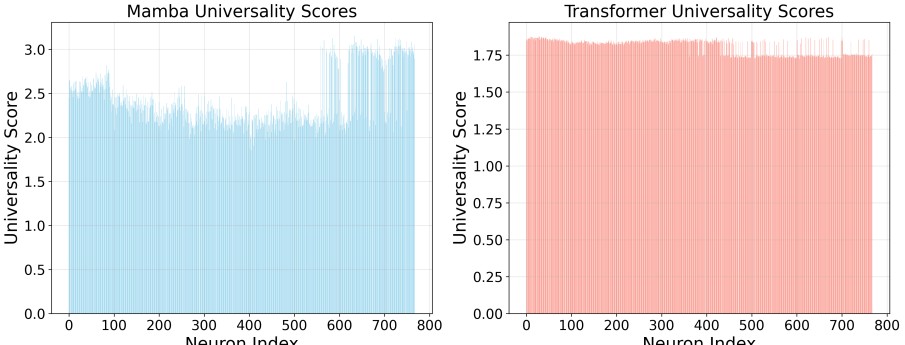

*Figure A1. Universality score distribution of subspaces.* Mamba vs. Transformer

## A.2. Causal subspaces

### A.2.1. INFLUENCE OF CAUSAL ACTIVATION

We observe from Fig. A3 that Causal subspaces (Song et al., 2024) although having the mean activation values in Mamba are significantly smaller that the Transformer, the cumulative impact indicates that subspaces in both models affects the output similarly. Additionally, the cumulative results reiterates that Mamba with a deep curvature relies on a small set specialised subspaces compared to the distributed subspace influence indicated by a flatter Transformer curve.

Cross-layer causal influence in Fig. A4 shows that early layers in Mamba exert large causal influence on later layers, consistent with their role in constructing and routing information. Within each layer, only a small fraction of activation subspaces exhibit large causal effects under intervention, while most remain functionally inactive. This indicates that Mamba's computation is driven by a sparse set of high-impact subspaces rather than a distributed population of units.

**Mathematical Proof:** Causal attention in Transformers is defined as in Eq. (4) and satisfies the normalization property $\sum_{s=1}^{t} \alpha_{t,s}^{\text{GPT}} = 1$.

$$\alpha_{t,s}^{\text{GPT}} \frac{\exp\left(\frac{q_t^\top k_s}{\sqrt{d_k}}\right)}{\sum_{u=1}^{t} \exp\left(\frac{q_t^\top k_u}{\sqrt{d_k}}\right)} \tag{4}$$

The Transformer output at position $t$ is $\left|y_t^{\text{GPT}}\right| \left|\sum_{s=1}^{t} \alpha_{t,s}^{\text{GPT}} v_s\right| \leq \sum_{s=1}^{t} \alpha_{t,s}^{\text{GPT}} |v_s|$, where the inequality follows from the triangle inequality. Assuming the value vectors have comparable norms across tokens, $|v_s| \approx |v|$, we simplify $|y_t^{\text{GPT}}| \approx |v| \sum_{s=1}^{t} \alpha_{t,s}^{\text{GPT}} |v|$.

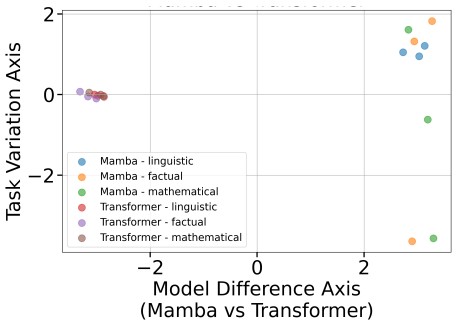

*Figure A2. 2D PCA Projection of Universality subspaces.* Mamba representations are less tightly clustered than Transformer representations. PC1 corresponds to task type; PC2 corresponds to model type (Mamba vs. Transformer)

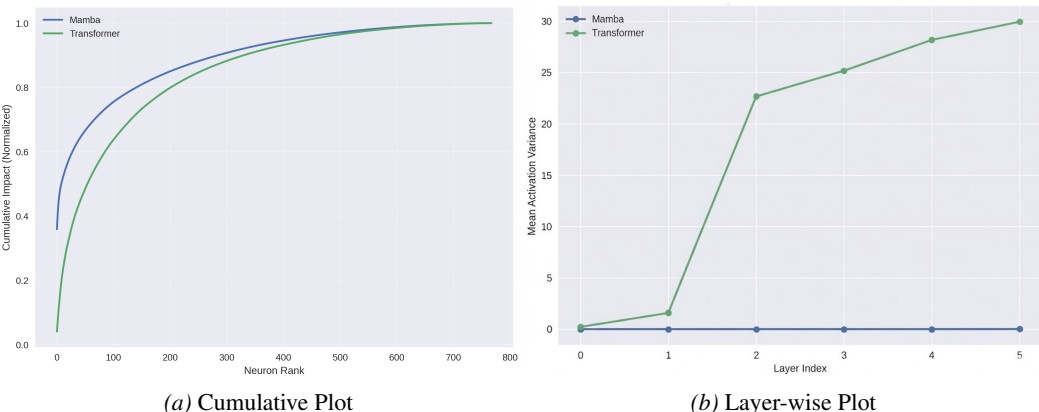

*(a)* Cumulative Plot        *(b)* Layer-wise Plot

*Figure A3. subspace-level causal influence in Mamba and Transformer.* (a) Cumulative causal impact distribution across subspaces. (b) Layer-wise activation contributions across the network.

In Mamba, the attention weight is defined by Ali *et al.* (Ali et al., 2025b) given bt Eq. 5.

$$\alpha_{t,s}^{\text{SSM}} = \sum_{m=1}^{N} Q_t^{(m)} H_{t,s}^{(m)} K_s^{(m)} \tag{5}$$

where,
s = source token position,
t = predicted token position,
$Q_t^{(m)} = S_C(\hat{x}^t)[m]$, $K_s^{(m)} = \text{softplus}(S_\Delta(\hat{x}^s)[m]) \, S_B(\hat{x}^s)[m]$,
$H_{t,s}^{(m)} = \exp\left(a_m \sum_{k=s+1}^{t} \text{softplus}(S_\Delta(\hat{x}^k)[m])\right)$,
$v_s = x_s$

Rewriting Eq. (5) using $C_t^{(m)} = S_C(\hat{x}^t)[m]$, $\Delta_s^{(m)} = \text{softplus}(S_\Delta(\hat{x}^s)[m])$, and $B_s^{(m)} = S_B(\hat{x}^s)[m]$, we represent $\alpha_{t,s}$ by Eq. 6.

$$\alpha_{t,s}^{\text{SSM}} \propto C_t^{(m)} \exp\left(a_m \sum_{k=s+1}^{t} \Delta_k^{(m)}\right) \Delta_s^{(m)} B_s^{(m)} \tag{6}$$

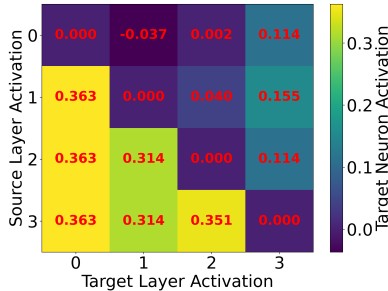

*Figure A4. Cross-Layer Causal Influence in Mamba.*

Empirically, these quantities satisfy

$$C_t^{(m)} \sim \mathcal{O}(10^{-1} - 1), \tag{7}$$

$$\Delta_s^{(m)} \sim \mathcal{O}(10^{-2} - 10^{-1}), \tag{8}$$

$$B_s^{(m)} \sim \mathcal{O}(10^{-1} - 1), \tag{9}$$

$$a_m < 0 \tag{10}$$

Since $a_m < 0$ and $\Delta_k^{(m)} > 0$, the history term satisfies $0 < H_{t,s}^{(m)} \leq 1$ and converges to zero as $t - s \to \infty$. Thus, long-range contributions decay exponentially with token distance.

Since Transformer attention weights are normalized while SSM attention weights are exponentially damped by the history term and therefore not normalized, the relative output magnitude can be approximated by the total SSM attention as shown in Eq. 11. This indicates that Mamba outputs have substantially smaller magnitude than Transformer outputs, thus proving the validity of magnitude difference observed in Fig. A3b.

$$\frac{\|y_t^{\text{SSM}}\|}{\|y_t^{\text{GPT}}\|} = \frac{\sum_{s=1}^t \alpha_{t,s}^{\text{SSM}}}{1} \approx \sum_{s=1}^t \alpha_{t,s}^{\text{SSM}} \tag{11}$$

### A.2.2. SPARSITY ANALYSIS

Sparsity analysis in Fig. A5 is plotted by computing the Gini index over subspace-wise activation variance, which captures inequality in activation distributions and measures sparsity. The cut-off for determining sparse subspaces is derived from cumulative variance threshold plots shown in Fig. A6, where the threshold is selected at the point where cumulative contribution saturates. Using this criterion, sparsity thresholds of 1.2 for Mamba and 1.5 for the Transformer are identified.

Fig. A5 shows that Mamba exhibits high sparsity of 60% in early layers that steadily decreases with depth to 15%, indicating selective filtering followed by progressive information integration due to recurrence as discussed in Section A.8. In contrast, Transformer has 55% sparsity which is lower than Mamba and decreases rapidly to 15%, resulting in more distributed activations across layers.

As shown in Table 1, Mamba has high $\Delta$PPL compared to Transformer. This behavior is studied using three complementary methods, i.e., gradient sensitivity, activation variance and Lipschitz constant (Scaman & Virmaux, 2018) measuring the propagation of perturbations through the network. Gradient sensitivity quantifies the effect of small perturbations on hidden states in terms of output logits, activation variance captures output instability across varying inputs and the Lipschitz constant measures worst-case amplification across layers.

Results show that Mamba is significantly more sensitive to perturbations than GPT-2 across all metrics. Mamba's average gradient sensitivity is 75.73±54.02, compared to 17.57±7.81 for GPT-2. This shows 6.61 times increase in early layers. Similarly, Mamba's average maximum Lipschitz constant is 3522.67±6964.23, which is 25.71 times higher than GPT-2 with 251.46±176.34. Further, the gradient values decay with depth thus resulting in higher $\Delta$PPL and influencing downstream behavior. In contrast, GPT-2 has comparatively stable sensitivity across layers, indicating lower $\Delta$PPL and more evenly distributed computation.

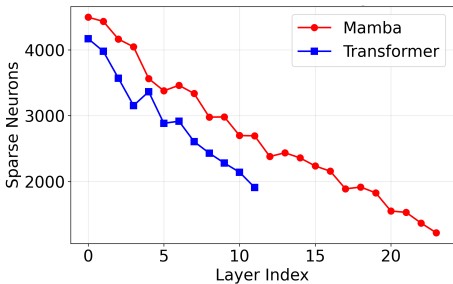

*Figure A5. Layer-wise subspace sparsity using Gini Index.* Mamba vs. Transformer

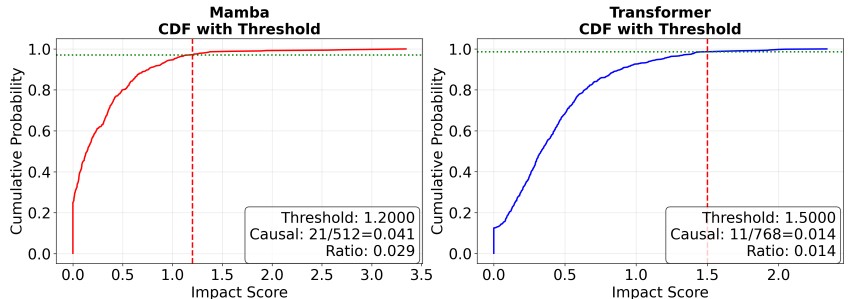

*Figure A6. Cumulative activation variance determining sparsity thresholds in Mamba and Transformer*

## A.3. Delta-Sensitive subspaces

Further results supporting centralised storage in Mamba vs decentalised storage in Transformer for bias inputs are show in Fig. A7.

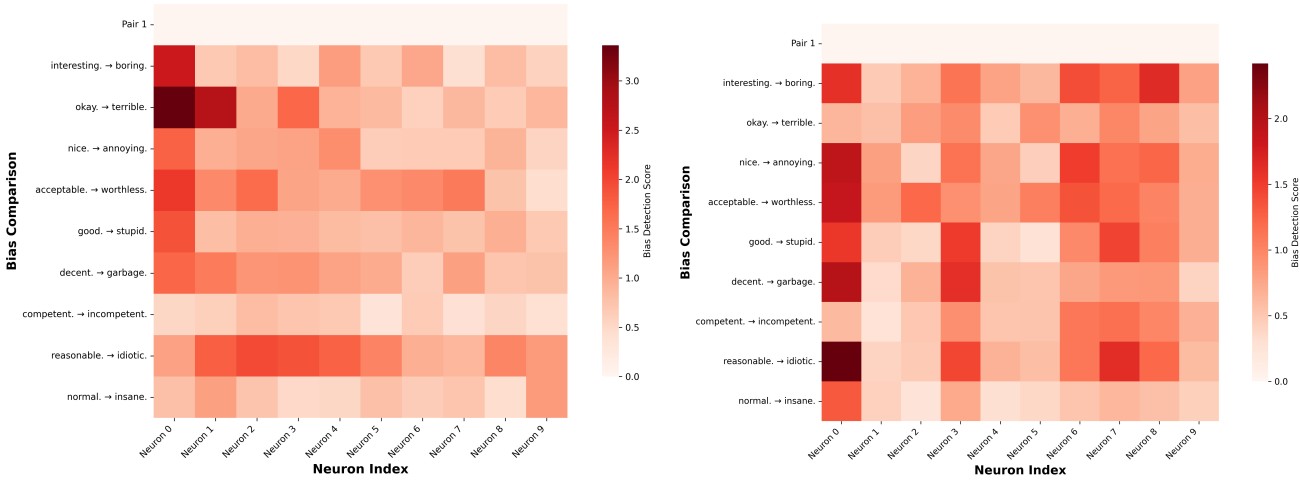

*(a)* Centralized bias sensitivity in a tightly coupled subspace cluster in Mamba-130M

*(b)* Distributed bias sensitivity across weakly coupled subspaces in GPT-2

*Figure A7. Bias heatmaps for delta-sensitive subspaces.* Mamba vs. Transformer

## A.4. Knowledge subspaces

Table A1 reports knowledge subspace overlap across facts using Jaccard similarity and intersection size for intra and inter-relations. Both models show high overlap in early layers and decreasing overlap with depth, indicating a shift from shared to specialized knowledge. Mamba with higher Jaccard preserves consistency across layers better than Transformers with Jaccard drastically reducing with depth, making early-to-middle layers more suitable for knowledge extraction.

*Table A1. Knowledge subspace overlap statistics for the topic: "Capital of France"*

*(a)* Mamba

| Layer | Relation | #Pairs | Avg Jaccard | Avg ∩ Size |
|---|---|---|---|---|
| 0 | Intra | 9 | 0.9693 | 744.33 |
| 0 | Inter | 57 | 0.9754 | 749.05 |
| 7 | Intra | 9 | 0.9592 | 736.22 |
| 7 | Inter | 57 | 0.9637 | 739.93 |
| 14 | Intra | 9 | 0.9389 | 720.00 |
| 14 | Inter | 57 | 0.9504 | 729.30 |
| 21 | Intra | 9 | 0.8972 | 681.11 |
| 21 | Inter | 57 | 0.8966 | 684.86 |
| 23 | Intra | 9 | 0.7169 | 483.00 |
| 23 | Inter | 57 | 0.6480 | 468.12 |

*(b)* Transformer

| Layer | Relation | #Pairs | Avg Jaccard | Avg ∩ Size |
|---|---|---|---|---|
| 0 | Intra | 9 | 0.9416 | 720.67 |
| 0 | Inter | 57 | 0.9463 | 725.77 |
| 4 | Intra | 9 | 0.7517 | 519.78 |
| 4 | Inter | 57 | 0.6939 | 507.40 |
| 8 | Intra | 9 | 0.7533 | 518.22 |
| 8 | Inter | 57 | 0.6979 | 508.35 |
| 11 | Intra | 9 | 0.3813 | 76.11 |
| 11 | Inter | 57 | 0.2306 | 59.84 |

### A.5. Dead subspaces

Dead subspaces are those that remain inactive across a large and diverse dataset (Voita et al., 2024). Fig. A8a shows that the percentage of dead subspaces decreases with depth, indicating that earlier layers contain more inactive subspaces, while later layers utilize representational capacity more effectively.

Fig. A8b presents the distribution of maximum activation per subspace. Dead subspaces correspond to very low maximum activation values, whereas the prominent peak around 1.5 indicates that most subspaces are actively engaged. Only a small fraction exhibit consistently low activation.

Fig. A8c shows activation frequency across inputs, capturing recurrence. This complements Fig. A6, which measures activation magnitude. A subspace may reach high activation but rarely fire; thus, considering both magnitude and frequency provides a more complete characterization of dead subspaces.

### A.6. Token-Detection subspaces

Token-detection subspaces activate reliably in the presence of specific input tokens. For each sequence, we record token-level activations and classify a subspace as token-detecting if it consistently shows high activation for a particular token (e.g., 'the', 'apple', punctuation). This identifies subspaces specialized for lexical pattern recognition.

Fig. A9a shows the number of token-detecting subspaces across layers. Counts peak in early layers, reflecting low-level pattern processing and decline with depth as representations become more abstract. Larger Mamba models (1.4B, 2.8B) accumulate token-detecting subspaces more strongly in middle layers, while smaller models saturate earlier. GPT-2 maintains a relatively low and stable count across depth.

Fig. A9b presents cumulative token coverage. Smaller Mamba models achieve near-complete coverage in early layers, whereas larger models require greater depth. GPT-2 shows high coverage from initial layers, indicating more distributed token encoding.

Fig. A9c shows newly introduced tokens per layer. GPT-2 introduces most tokens in the first layer, with minimal growth afterward. Smaller Mamba models add tokens primarily in early layers, while larger models continue adding new tokens through middle layers, suggesting a more gradual allocation of token recognition across depth.

### A.7. N-Gram subspaces

N-Gram subspaces activate for a specific sequences of N tokens (N-grams) rather than individual tokens (Voita et al., 2024) to encode common phrases and multi-word patterns. They are identified by measuring the correlation between subspace activations and specific N-grams (e.g., 'in the end', 'as a result', 'for example'), with strongly correlated subspaces classified as N-gram subspaces.

From Fig. A10, Mamba exhibits hierarchical processing, with early layers performing localized N-gram feature detection and deeper layers integrating these into abstract representations, reflecting recurrent state and selective mechanisms. In contrast, GPT-2 shows more uniformly distributed subspace specificity across layers due to global attention, resulting in generalizt representation.

Further, we observe a systematic shift in the depth at which N-gram detection emerges as model size increases. In smaller

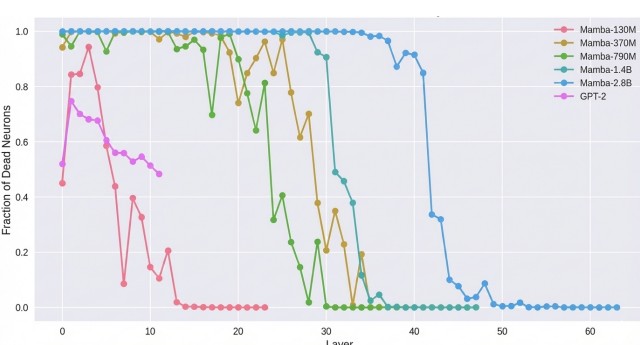

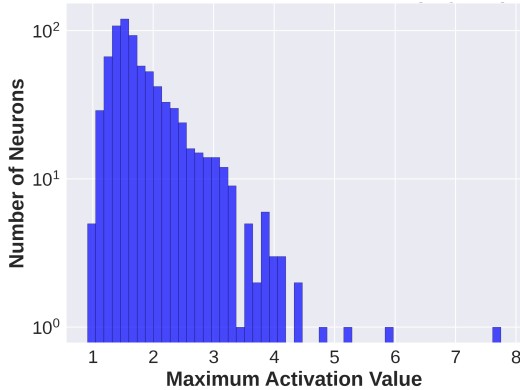

*(a)* Percentage of dead subspaces across layers for different models

*(b)* Distribution of the maximum activation value for each subspace

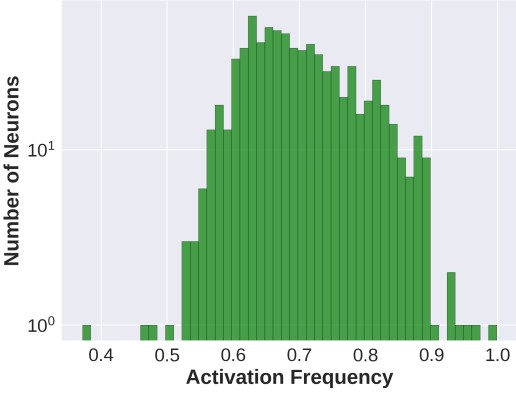

*(c)* Activation frequency of subspaces

*Figure A8. Dead subspace analysis across Mamba model variants and Transformer*

Mamba models, N-gram subspaces emerge majorly in early layers. In contrast, larger Mamba models exhibit a shift in N-gram detection toward middle layers. This indicates that subspace specialisation for higher-order patterns is dependent on model capacity. Larger models construct richer and distributed representations in early layers and specialises as depth increases. Smaller models, constrained by limited capacity, specialize earlier and directly encode N-gram patterns through sparse activations, thus showing a size-dependent shift as seen in Fig. A11. In GPT, the attention mechanism supports simultaneous specialization across layers resulting in parallel specialization strategy that is invariant to model size.

### A.8. Recurrence Analysis

Mamba uses gated recurrent updates to propagate information across depth. Fig. A12a analyzes the temporal dynamics of recursive memory using temporal variance and cross-layer activation changes. We observe that temporal variance is high in early layers, drops sharply in mid-layers consistent with selective SSM gating and gradually increases in deeper layers. This shows accumulation of structured temporal representations through recursive state updates. Cross-layer activation variation seen in Fig. A12b, highlights the magnitude of feature transformation across depth, with larger changes in early transitions showing rapid state updates and convergence in long-range transitions indicating cumulative state integration.

**Cross-layer correlation:** Cross-layer projections reveal sparse but highly selective inter-layer connectivity. As demonstrated in Fig. A13, mean correlation values remain extremely low ($0.0015 \pm 0.0001$), indicating that most activation dimensions evolve independently. In contrast, maximum correlations are consistently high ($0.9981 \pm 0.0010$), showing that a limited subset of dimensions transmits critical information across layers. This pattern is stable across layer pairs (e.g., Layer $0 \rightarrow 1$: mean 0.0016, max 0.9968; Layer $1 \rightarrow 2$: mean 0.0015, max 0.9991), demonstrating targeted recursive projection rather than uniform transformation.

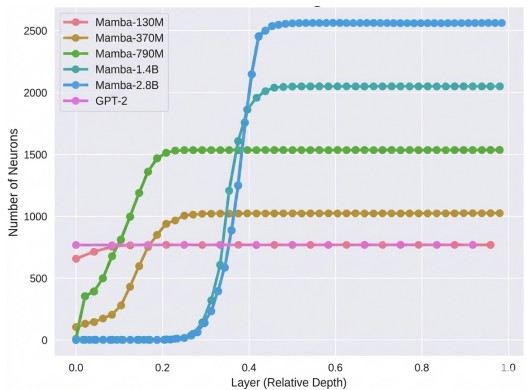

*(a)* Number of token-detecting subspaces across layers for Mamba models and GPT-2

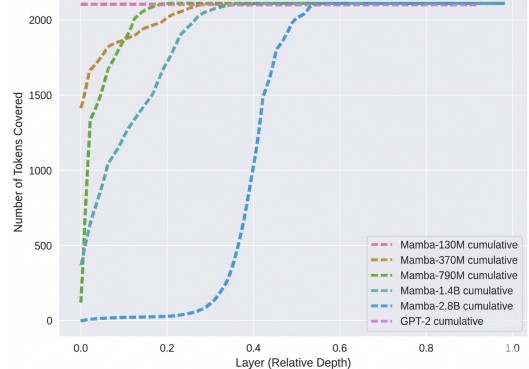

*(b)* Cumulative unique tokens covered by token-detecting subspaces

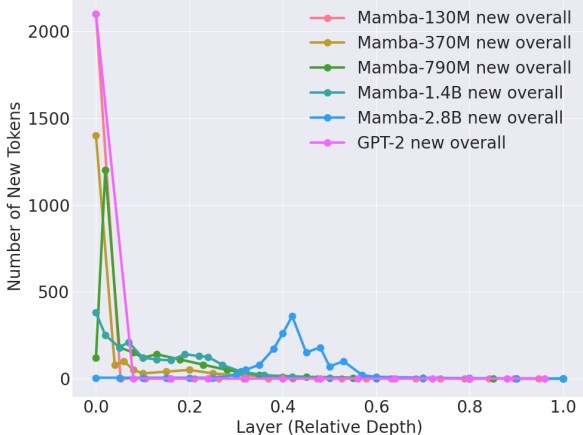

*(c)* Number of newly introduced tokens detected at each layer

*Figure A9. Layer-wise dynamics of token-detecting subspaces across Mamba model variants and Transformer.* Highlights size-dependent differences in subspace allocation, token coverage and the depth at which new token representations emerge

**Temporal dynamics:** Temporal variance exhibits a clear three-stage pattern as seen in Fig. A12a, early layers show high variance (Layer 0: $\sim 0.102$), reflecting comprehensive input processing. Variance drops sharply in mid-layers (Layer 2: $\sim 0.012$) due to selective filtering by state-space gating. Deeper layers show gradual variance recovery (up to $\sim 0.050$ by Layer 10), indicating accumulation of complex temporal structure through compounding recursion.

**Long-range dependency integration** As shown in Fig. A12b, normalized recursive and activation changes closely align across all layer transitions. Short-range transitions exhibit modest increase in recursiveness of 3.4% to 4.3% as seen in Layer $0 \rightarrow 3$, $0 \rightarrow 6$, corresponding to early state construction. In contrast, long-range transitions show near-saturation, with recursive changes reaching 99.7% to 100.0% and activation changes 99.0% to 99.2%, as seen in layer transitions $0 \rightarrow 23$, $3 \rightarrow 23$, $6 \rightarrow 23$, indicating cumulative integration across depth. The convergence of these high-percentage rises suggests stabilization of representations and long-range dependency consolidation.

### A.9. Induction Head Analysis

Induction head analysis examines how recursive state updates encode contextual information for pattern completion in repeated sequences. To assess induction in Mamba, we evaluate its ability to reproduce repeated subsequences by computing cosine similarity between recurrent states at matching positions.

Mamba shows high similarity between matched states, indicating effective repeated-pattern recognition. In contrast to transformers, which enable immediate token-level copying via attention (Olsson et al., 2022), Mamba requires longer

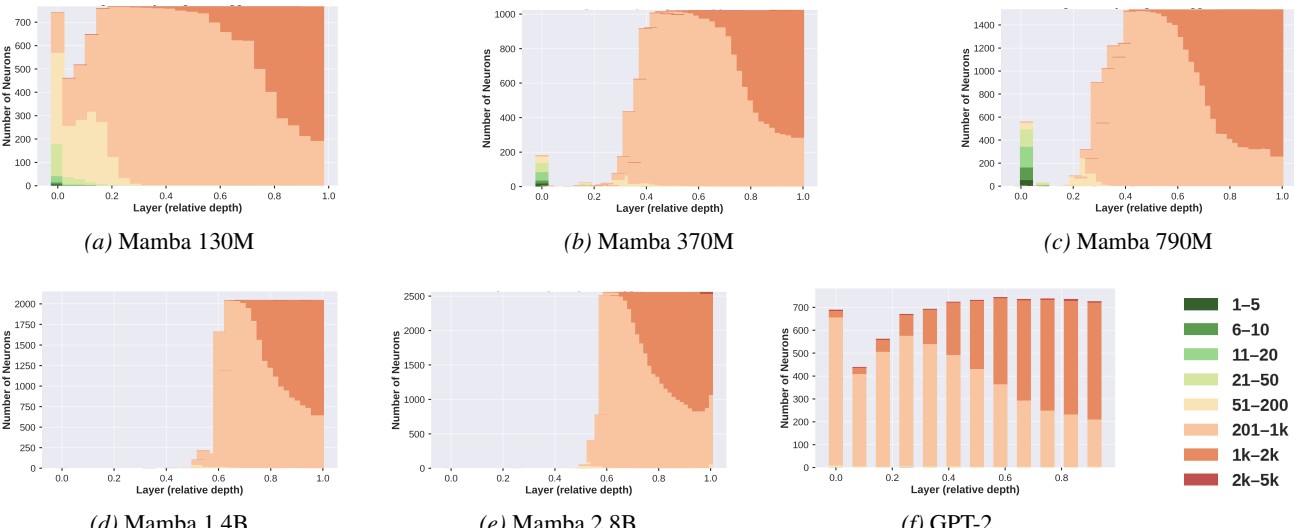

*Figure A10. N-Gram subspaces across layers and model scales*. Each plot shows the number of subspaces in each layer that respond to unique unigrams, grouped by specificity. Dark green bins indicate highly specific subspaces, lighter green/yellow indicate moderately specific subspaces and orange/red indicate generalizt subspaces. GPT-2 exhibits relatively consistent N-Gram detection patterns across scales, reflecting attention-enabled parallel specialization. In contrast, Mamba models show a size-dependent shift in N-Gram selectivity, consistent with hierarchical recursive buildup through sequential state-space processing

contexts for induction to emerge. As contextual information is integrated incrementally through state-space recursion (Section A.8), induction strength increases with repeated context, as shown in Fig. A14, where growing hidden-state norms reflect cumulative recursive integration.

## B. Extended Mechanistic Interpretability Analyzes

In this section, we present additional mechanistic interpretability analyzes that complement the main results discussed in the paper. A brief overview of the interpretability methods, their descriptions and key findings is provided in Table A1.

### B.1. Hypothesis Probing

Hypothesis probes reveal that task-relevant information is concentrated in a small subset of SAE features with sparse, high activations and structured positional selectivity shown in Fig. B1. Task-relevant and architecture-specific circuits identified by grouping dimensions encoding task-relevant signals are listed below.

- SAE-based (Strength 0.87) - Derived from SAE features. Has strongest task alignment. Provides reliable and predictive representations (Wang et al., 2024b)
- Probe-based (Strength 0.82) - Identified via linear probe correlations. Captures predictive structure while underestimating nonlinear interactions
- SSM-specialised (Strength 0.50) - Architecture-specific features reflecting state-space mechanisms
- Temporal (Strength 0.40) - Capture sequence dynamics that are weakly correlated with task labels

### B.2. Polysemanticity

We evaluate the degree of feature superposition and polysemanticity by analysing interference, effective dimensionality and activation sparsity within individual layers. From Fig. B2a, We observe a mean interference score of 0.58, indicating mild superposition, where features partially overlap within shared dimensions rather than being severely entangled.

The effective dimensionality plotted in Fig. B2b is 0.189 which is much lower than the hidden size. This shows that the model reuses a limited subset of dimensions to encode a larger number of features. The compression property supports the superposition hypothesis, indicating feature sharing across subspaces. Additionally, activation sparsity is 0.015, reflecting dense activations.

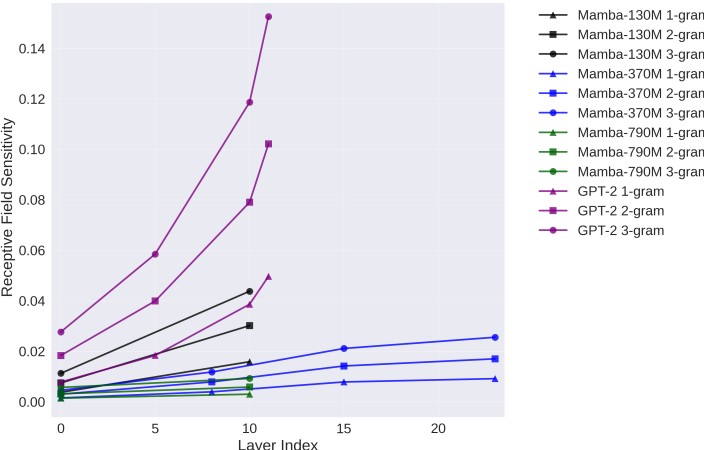

*Figure A11. Receptive field sensitivity across layers for 1-gram, 2-gram and 3-gram in Mamba model variants and Transformer.* Larger Mamba models exhibit a delayed emergence of higher order N-gram sensitivity while GPT-2 shows strong, depth-invariant specialisation

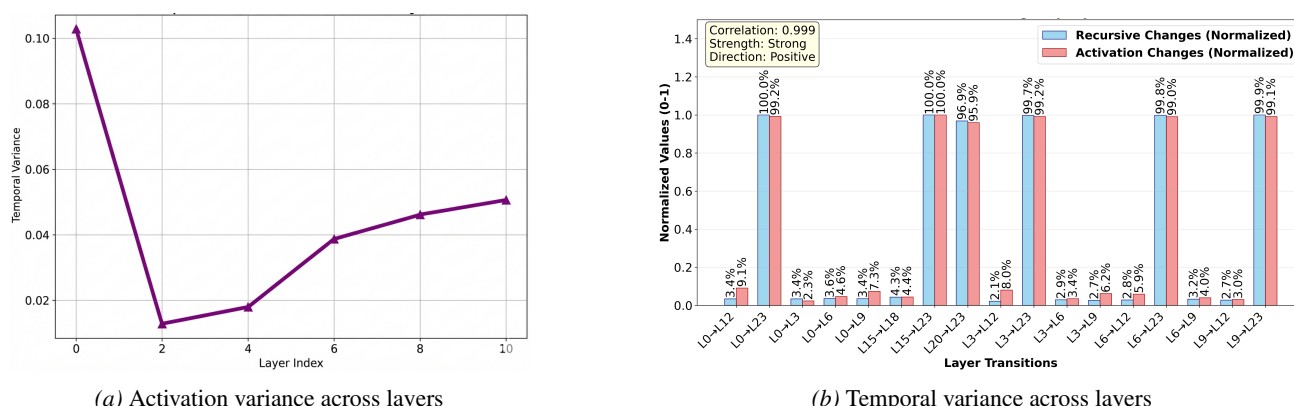

*(a)* Activation variance across layers        *(b)* Temporal variance across layers

*Figure A12. Cross-layer recursive dynamics in Mamba.* (a) Temporal variance across layers shows a three-stage pattern: high variance in early layers (Layer $0 \approx 0.102$), a sharp reduction in mid-layers (Layer $2 \approx 0.012$) consistent with selective SSM gating and a gradual increase in deeper layers (Layer $10 \approx 0.05$) indicating accumulation of structured temporal representations through recursion. (b) Cross-layer activation changes measure the magnitude of feature transformation between layers, with larger changes in early transitions reflecting rapid state updates and convergence in long-range transitions indicating cumulative state integration across depth

Together, these results show that the model efficiently utilises its dimensions while maintaining sufficient separation between features to preserve interpretability. Although subspaces are polysemantic, individual features remain distinguishable despite shared representational space.

## B.3. Grokking

We analyze training dynamics in Mamba using grokking by characterizing the transition from memorisation to generalization and identify feature-level triggers for shift (Tlaie, 2024) as seen in Fig. B3a. The analysis tracks feature emergence over training, detects abrupt performance changes and measures representation stability.

A clear phase transition occurs at epoch 132, where performance increases sharply by 0.18, marking the onset of generalization. Critical token at positions 42, 156 and 203 emerge sequentially at epochs 125, 128 and 130 as seen in Fig. B3b. Final feature appears before the performance jump, indicating a continuous activation buildup that triggers generalization.

Table B2, emphasise these results by showing that only 10% of features are used in generalization, exhibiting moderate stability and low co-activation. Across training, average values are $7.55 \pm 1.31$ with 17% CoV. The dynamics follow a cyclic pattern with a period of approximately 44 steps, reflecting oscillation between exploration and exploitation. Overall, Mamba exhibits smooth and continuous learning dynamics with a dominant generalization transition, in contrast to transformer

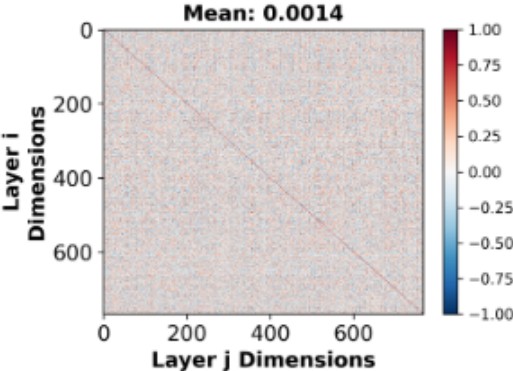

*Figure A13. Cross-layer activation correlations between consecutive network layers.* Mean correlations are near zero while maximum correlations approach unity, indicating sparse but highly selective inter-layer information transfer

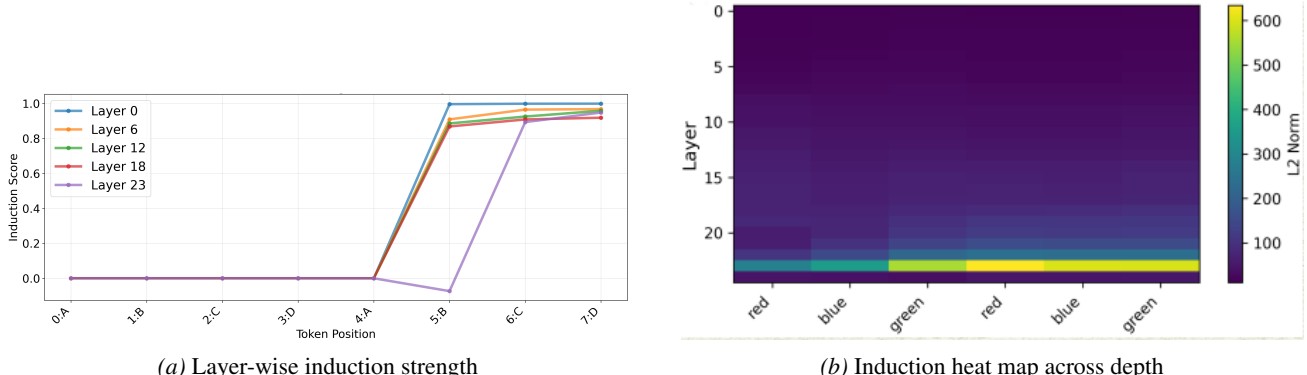

*(a)* Layer-wise induction strength          *(b)* Induction heat map across depth

*Figure A14. Induction behavior in Mamba*

models that display multiple abrupt grokking events.

### B.4. Causal Validation using Activation Patching

Targeted activation interventions are used to quantify the causal requirement and self-sufficiency of the identified circuits. For each circuit, we perform activation patching, where activations corresponding to the circuit dimensions are copied from a reference input and injected into a test input at the same layer. We measure the resulting change in the target output distribution relative to the unpatched baseline. Further, we ablate circuits by zeroing the activations and measure the performance degradation.

We observe that SAE-based circuits have the strongest causal influence with causal effect size, necessity score and sufficiency score of 0.65, 0.72 and 0.58 respectively. Probe circuits show weaker effect with causal effect, necessity and sufficiency of 0.48, 0.55 and 0.42 respectively, consistent with their lower circuit strength. Sufficiency scores remain low for all circuit types, indicating that no individual circuit is sufficient in isolation. Hence, Mamba's behavior builds from cooperative interactions among multiple circuits, supporting a distributed rather than modular computational organization.

### B.5. Dynamic Universality of Circuits

To assess the generalization of the discovered Mamba circuits across domains, we measure activation consistency across syntactic, semantic, narrative and code-oriented inputs at the circuit level. We compute a universality score (Gurnee et al., 2024) of 78%, with high cross-domain consistency of 0.7–0.8 on linguistic tasks, indicating that Mamba learns domain-general representations through coordinated, distributed computation.

In contrast, code-oriented inputs exhibit lower consistency and temporal stability of 0.45 and 0.58, respectively, suggesting

| Method | Description | Key Findings |
|---|---|---|
| Sparse Autoencoders (SAEs) | Reconstruct hidden activations using sparse latent representations to identify activation-level features and compact summaries of model activations | Identified sparse task-relevant features carrying most of the useful signal. SAE analysis revealed extreme sparsity and emergent specialized features responsible for key behaviors |
| Dictionary Learning | Applied on SAE latent representations to decompose activations into interpretable components and quantify sparsity, reconstruction error and feature utilization | Showed that sparsity, reconstruction error and feature usage occupy distinct latent dimensions, supporting functional specialization within representations |
| SPD (Stochastic Parameter Decomposition) | Decomposes parameters into sparsely used vectors and computes entropy, variance, effective rank, gradient sensitivity and post-ablation KL divergence | Identified a major Layer 20 information bottleneck characterized by high entropy, high variance and extreme KL divergence after ablation |
| Hypothesis Probing | Probe-based analysis examining whether task-relevant information is concentrated in sparse SAE-discovered features and circuits | SAE-based circuits showed strongest task alignment and predictive structure with structured positional selectivity |
| Polysemanticity Analysis | Measures feature interference, effective dimensionality and activation sparsity to analyze feature superposition | Found moderate superposition and low effective dimensionality, indicating substantial feature reuse across dimensions |
| Grokking Analysis | Analyzes emergent feature organization, induction behavior and transition dynamics during training | Observed sparse feature specialization, gradual learning dynamics and increasing induction strength during recursive integration |
| Causal Validation using Activation Patching | Copies, amplifies or ablates discovered circuit activations to evaluate causal necessity and sufficiency | SAE-based circuits showed strongest causal influence, supporting distributed rather than modular computation |
| Dynamic Universality Analysis | Measures cross-domain activation consistency and circuit generalization across linguistic and code tasks | Found strong universality across linguistic tasks but weaker consistency for code tasks, indicating specialized sequence-processing dynamics |
| Scaling Analysis | Compares feature specialization, N-gram subspaces and induction behavior across different model scales | Observed scale-dependent shifts in feature selectivity and hierarchical recursive buildup in Mamba models |
| APD (Attribution Path Decomposition) | Combined with SPD to decompose computation into operational phases and functional modules | Identified five operational phases including a critical Layer 20 bottleneck controlled by $dt\_proj.bias$ |

*Table A1.* Summary of mechanistic interpretability methods, analyses and key findings

*Table B2.* Summary of grokking-related training metrics.

| Metric | Value | Interpretation |
|---|---|---|
| Emerging features | 77/768 (10%) | Sparse specialization |
| Grokking score | 0.023 | Gradual learning |
| Transition points | 58 | Frequent micro-shifts |
| Average stability | 0.163 | Moderate consistency |
| Gradient range | 5.4 - 14.0 | Stable convergence |
| Feature coherence | 0.15 | Low co-activation |

reliance on specialized features and distinct sequence-processing dynamics. This divergence indicates that token-level dependencies and long-range control flow require specialized circuits rather than general linguistic mechanisms. Together, these results show that while individual dimensions are highly specialized, generalization occurs due to interactions among groups of subspaces forming stable circuits.

Comparing Mamba and transformer circuits shows low architectural equivalence of 0.57, despite high activation similarity of 0.78 and low functional similarity of 0.23. Only 5% of features are transferable between architectures, indicating that superficially similar features often encode different information and that architectural differences influence function representation.

## B.6. Scaling Analysis

Scaling laws are derived across Mamba models to assess the evolution of circuits with capacity. We observe that the core mechanisms are largely scale-invariant, with 78% of features preserved across model sizes and cross-scale similarity ranging from 0.75 to 0.82, indicating stable underlying computations. Circuit complexity increases linearly with model size, indicating the emergence of additional specialized circuits with 14% increase in features in larger models. This suggests that scaling enables new capabilities while preserving core interpretability.

We identify bottleneck across model scales, occurring at a layer depth ratio is $85.4\% \pm 2.1\%$ as shown in Table. B3. This

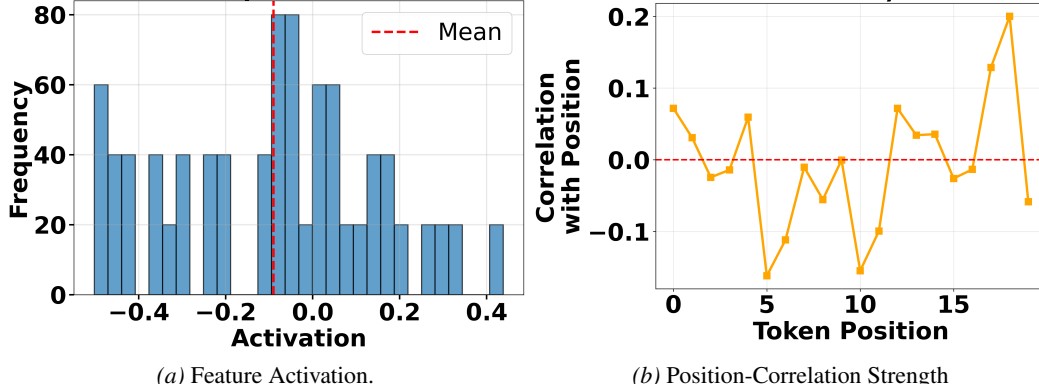

*(a)* Feature Activation.                    *(b)* Position-Correlation Strength

*Figure B1. Hypothesis probing of SAE-discovered features.* (a) Activation strength distribution for a representative SAE feature ($\rho = 0.479$) follows a sparse, power-law pattern, where most activations are near zero and a small subset carries task-relevant signal, (b) Correlation between SAE features and token position varies non-uniformly across the sequence, peaking at mid-sequence positions (Layers 15–18), indicating recurrence and structured positional selectivity

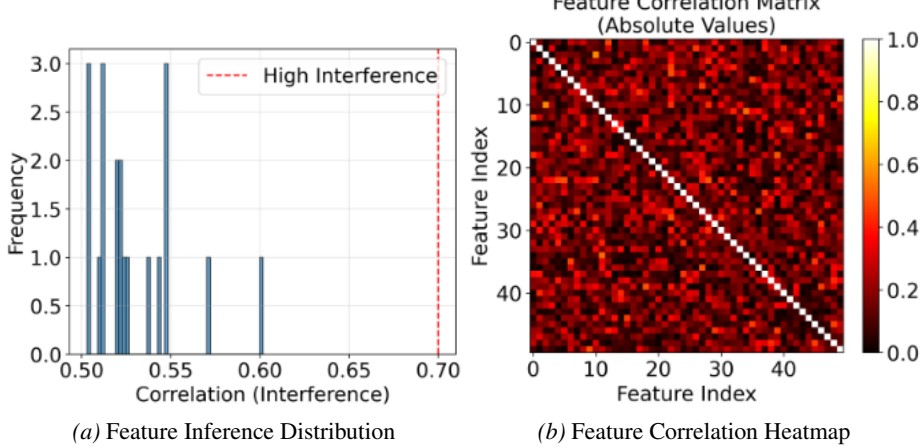

*(a)* Feature Inference Distribution            *(b)* Feature Correlation Heatmap

*Figure B2. Polysemanticity in Mamba*

scale-invariant bottleneck suggests an architectural regularity in how Mamba allocates representational capacity.

*Table B3. Scale-invariant bottleneck location across Mamba model variants*

| Model | Layers | Bottleneck Layer | Depth Ratio (%) |
|---|---|---|---|
| Mamba-130M | 24 | 20 | 87.5 |
| Mamba-370M | 48 | 40 | 83.3 |
| Mamba-790M | 48 | 41 | 85.4 |

Further, we observe a trade-off between modularity and circuit complexity as model size increases as recorded in Table B4. Smaller models exhibit higher modularity with more distributed computation, whereas larger models show reduced modularity and increased circuit complexity, indicating a structured allocation of resources over global sharing.

## B.7. Further Experiments to Validate the Bottleneck at Layer 20

To validate the presence and the effect of information bottleneck at Layer 20, we conduct additional analyzes examining entropy–variance dynamics, parameter-level training behavior and non-linear gating characteriztics.

### B.7.1. ENTROPY-VARIANCE PHASE TRANSITION

We observe a clear entropy–variance phase transition at Layer 20. As shown in Table B5, entropy increases from Layer 19 to Layer 20 before sharply decreasing at Layer 21, indicating an expansion–compression pattern where information

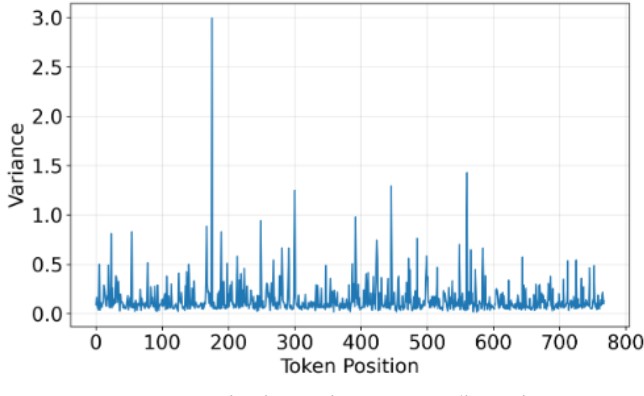
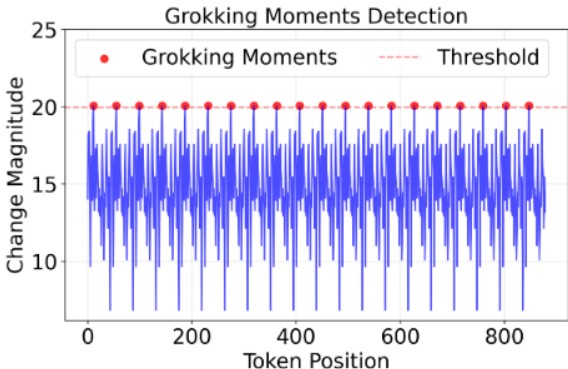

*(a)* Feature activation variance across dimensions

*(b)* Detection of grokking moments during training

*Figure B3. Grokking dynamics in Mamba.* The model exhibits structured feature emergence and a discrete generalization transition following stable memorization

*Table B4. Modularity–complexity trade-off across Mamba model variants*

| Model Size | Modularity | Circuit Complexity |
|---|---|---|
| 130M | 1.0061 | 1.0 |
| 370M | 1.0028 | 2.0 |
| Correlation | −1 | +1 |

is first explored and then selectively committed. Further, variance increases monotonically across these layers, reflecting growing representational selectivity. Effective rank peaks at Layer 20 and then reduces, further supporting a transition from distributed exploration to compressed representation.

*Table B5. Entropy–variance dynamics around the bottleneck layer*

| Layer | Entropy | Variance | Effective Rank |
|---|---|---|---|
| 19 | 1.02 | 2.30 | 6.25 |
| 20 | 1.19 | 2.63 | 7.59 |
| 21 | 0.92 | 3.59 | 6.49 |

### B.7.2. FROZEN MASTER GATE DURING TRAINING

At Layer 20, *dt_proj.bias* behaves as a master gating controlling the SSM time constant ($\Delta t$). This parameter exhibits near-frozen behavior during training, with gradients approximately 11x smaller than other parameters in the same layer. CoV being 0.001 is extremely low, indicating high stability. Ablation highlights performance degradation with KL divergence of 813 indicating that the parameter is frozen at an optimal value as seen in Table B6. These results conclude that *dt_proj.bias* establishes the bottleneck early in training and remains fixed thereafter, with the rest of the network adapting around it. This behavior can be expressed as a three-phase learning process,

1. Early training (epochs 1–5) - *dt_proj.bias* rapidly explores the parameter space to identify optimal gating threshold

2. Mid training (epochs 6–20) - Gradients reduce as the optimal value is reached

3. Late training (epochs 21+) - Parameter remains effectively frozen while downstream representations adapt

**Consistency across neighboring layers.** To verify that the Layer 20 master gate behavior is not a result of a single measurement instance, we examine attribution magnitudes (Braun et al., 2025) for dt_proj.bias across adjacent layers (L19–L21) over multiple evaluation checkpoints:

$$\text{Attribution}_{\text{L19–L21}} = \begin{bmatrix} 0.0031 & 0.0055 & 0.0009 \\ 0.0050 & 0.0083 & 0.0010 \\ 0.0035 & 0.0074 & 0.0009 \end{bmatrix}$$

Rows correspond to independent evaluation checkpoints and columns correspond to Layers 19, 20 and 21 respectively. The small magnitudes across all entries indicate that the gate operates in a stable environment once training has progressed beyond initial feature extraction phase. However, Layer 20 consistently shows the highest attribution within each checkpoint, demonstrating that despite its frozen dynamics, it remains the dominant temporal control parameter relative to its neighboring layers.

*Table B6. Causal and stability metrics for* `dt_proj.bias` *at Layer 20*

| Metric | Value | Relative | Interpretation |
|---|---|---|---|
| APD Attribution | 0.00758 | 5.0x L19 | Causal impact |
| SPD Stability | CoV = 0.001 | 1000x avg | Parameter rigidity |
| Gradient Magnitude | 0.072 | 0.09x avg | Frozen dynamics |
| Ablation KL | 813.0 | 160x avg | Catastrophic loss |
| Prediction Δ | 45% | – | Output control |
| Susceptibility | -41.54 | 0.13 of max CoV | Perturbation resistance |
| BIF Sensitivity | 250.0 | Max cluster | Bifurcation trigger |

### B.7.3. NON-LINEAR GATING BEHAVIOR

Linear response analysis from dictionary learning (see Section 4.3.1 ) reveals a weak negative correlation of *r = -0.362* between parameter perturbation and output response, inconsistent with a linear gating mechanism. Sensitivity analysis further shows asymmetric responses to positive and negative perturbations, indicating non-linear behavior. Perturbation patterns follow sigmoid-like gating function, where *dt_proj.bias* determines a critical switching threshold.

The learned bias at Layer 20 regulates information flow with high specificity by decreasing the bias shifts inputs below the threshold and opens the gate, thus, increasing information flow. On the contrary, increasing the bias, suppresses inputs above the threshold and closes the gate. This behavior confirms that *dt_proj.bias* implements a non-linear, learned gating mechanism that enforces the information bottleneck and controls representational compression illustrated in Table B3.

### B.8. Detailed Mamba Decomposition into Operational Phases and Modules

Based on the SPD and APD (Braun et al., 2025) results in Section 4.3, we decompose Mamba's computation into five distinct operational phases, each characterized based on geometry, sparsity and causal sensitivity.

**Phase 1 - Feature Extraction (Layers 0–18):** Early layers construct rich, hierarchical representations while accumulating long-range dependencies in SSM states. Activations exhibit high entropy and low variance ($\sigma \approx 2.0 \pm 0.3$), indicating diverse but weakly differentiated features. SAE analysis reveals extreme sparsity (98.1% inactive; mean usage 1.88%), with 77 emergent features ($\sim$10% of layer capacity) carrying most task-relevant signal. This phase emphasizes broad feature discovery rather than specialization.

**Phase 2 - Pre-Bottleneck Compression (Layer 19):** Layer 19 prepares representations for reorganization, marked by an 18% increase in variance ($\sigma = 2.30$) while entropy remains stable ($H = 1.02$). The effective rank rises to 6.25, indicating increased dimensional engagement without information loss. This layer functions as a staging point for the subsequent bottleneck.

**Phase 3 - Information Bottleneck (Layer 20):** Layer 20 constitutes a sharp, input-dependent bottleneck governed by adaptive temporal gating, with *dt_proj.bias* acting as a master control parameter. Despite low gradient magnitude (0.072; $\sim$11$\times$ smaller than Layer 19) and extremely high stability (SPD CoV = 0.001), ablations induce catastrophic effects (KL divergence = 813; 45% prediction flips). Entropy peaks (+16%, $H = 1.19$), variance increases (+14%, $\sigma = 2.63$) and effective rank reaches its maximum (7.59), indicating maximal information reorganization rather than simple compression. This layer selectively filters and gates information critical for downstream computation.

**Phase 4 - Feature Decomposition (Layer 21):** Following the bottleneck, representations specialize for output alignment. Entropy drops sharply (–23%, $H = 0.92$), signaling completed compression, while variance surges (+36%, $\sigma = 3.59$). Effective rank decreases to 6.49, reflecting controlled dimensionality reduction and feature disentanglement.

**Phase 5 - Output Projection (Layers 22–23):** Final layers transform compressed representations into logit space. Layer 22 exhibits a dramatic variance explosion (+127%, $\sigma = 11.66$), amplifying discriminative features, while Layer 23 performs final vocabulary mapping. Activations show low entropy and extreme variance, consistent with sharply differentiated output representations.

Phases describe how computation evolves across depth but do not identify controllable parameters. To address this, we cluster parameters into five functional modules based on attribution strength, stability and causal influence. These modules connect layer-wise phases to parameter-level mechanisms, revealing Mamba's architectural organization and mechanistic bottlenecks. This decomposition provides insights beyond the phase-level view:

- Pinpoint control points: Modules identify the parameters responsible for key behaviors. For example, although Phase 3 exhibits a bottleneck, Module 2 shows that *dt_proj.bias* in Layer 20 (Temporal Gate) governs adaptive temporal dynamics, making it a high-leverage intervention point.
- Disentangle overlapping roles: Phases often combine multiple functions. In Phase 1 (Layers 0–18), modules separate state evolution (SSM Core), input-dependent transformations (Input Processing) and projection into the recurrent state (Input Projection).
- Link dynamics to causality: Modules clarify how parameters propagate or filter information. SSM Core parameters (*A_log*, *D*, *conv1d*) show moderate causal influence with input-dependent variation, while Output Projection weights behave as stable readouts.
- Enable targeted steering: Mapping phases to modules highlights which parameters most effectively alter behavior, allowing controlled changes to sequence dynamics and recurrent state evolution without disrupting unrelated computations.

**Module 1 - Output Projection (23 parameters):** Linearly maps residual stream to output space and stabilizes the model's global output scale. It consists of the output projection weights (*out_proj.weight*) across all layers with a mean effect of -41.45 and CoV of 0.13. Low CoV indicates that output projection behaves as a deterministic linear operator with consistent influence across inputs. This stability suggests that output scaling is largely decoupled from input-dependent dynamics and serves as a fixed readout mechanism.

**Module 2 - SSM Core (138 parameters):** Controls sequence processing and temporal state evolution. It is composed of the core state-space parameters *A_log*, *D*, *conv1d*, *x_proj* and *dt_proj* and has a mean effect of -41.35 with a moderate CoV of 0.45. A higher CoV indicates input-dependent temporal dynamics. Inclusion of stable (*A_log*) and adaptive components (*D* and *conv1d*) shows that Mamba has fixed state decay and flexible signal routing.

**Module 3 - Input Processing (41 parameters):** Mediates the transformation of raw tokens into internal representations and introduces input-sensitive adaptations. It comprises token embeddings, bias parameters and *conv1d* layer at Layer 7. The module shows a mean effect of -41.42 and CoV of 0.96. A high CoV indicates strong dependence on input content and token position, indicating high sensitivity to perturbations due to input contextualization and initialization of downstream dynamics.

**Module 4 - Input Projection (24 parameters):** Expands and gates representations entering the recurrent state space. Comprises input projection weights (*in_proj.weight*) across layers with a mean effect of -41.42 and CoV of 0.47. Module shows that input projection is more adaptive than output projection but less volatile than input processing indicating that the controlled expansion mechanism regulates information flow into the SSM core.

**Module 5 - Temporal Gate (2 parameters):** Exerts disproportionate control over temporal behavior at critical depths. Temporal gate comprises of *conv1d.bias* at Layer 0 and *dt_proj.bias* at Layer 20. Module has a mean effect of -41.54 and CoV of 0.11 and shows high causal influence and stability across inputs. This indicates a global temporal control function that controls sequence dynamics.

Interaction strength between modules is approximately zero, indicating Mamba combines functional contributions linearly rather than multiplicative or hierarchical gating.

**SPD-Based Subspace Clustering.** Having defined the constituent modules, we can further decompose Layer 20 using SPD attribution statistics (Section 3.1.2) to identify functional subspaces and discern the components actually affecting the performance. This yields the following eight KMeans clusters, where $k = 8$ is chosen as a balance between granularity and robustness. Nearby values produce similar groupings, with $k < 4$ merging distinct components and $k > 8$ fragmenting clusters without revealing additional structure.

- Cluster 1 (0.89): Output projection weights controlling activation magnitude

- Cluster 2 (0.76): SSM dynamics (A log, D, x proj, dt proj, conv1d)

- Cluster 3 (0.72): Layer normalization parameters (Chen et al., 2025a)

*Table B7. Causal Validation of Identified Mamba Circuits*

| Circuit Type | Intervention Type | Metric Affected | Effect Size | Interpretation |
|---|---|---|---|---|
| SAE-based Circuits | Feature Ablation | Perplexity | $\Delta$PPL $\approx$ +394.1% | Circuits are causally necessary for factual recall |
| Probe-based Circuits | Activation Amplification | Token Probability | $\Delta$Prob $\approx$ +218.0% | Circuits contribute directly to retrieval behaviour |
| Temporal Circuits | Information Gating ($\Delta$) | Accuracy | $\Delta$Acc $\approx$ -42% | Circuits control long-range memory and sequence integration |

• Clusters 4-8: Embeddings, biases, input/output projections, and fine-grained temporal and normalization parameters

### B.9. Causal Importance of Circuits Discovered through Mechanistic Interpretability

Causal intervention experiments validate the functional importance of these circuits by selectively perturbing their constituent components. Table B7 summarizes representative causal tests, illustrating the effects of targeted feature ablation, activation modulation and gating disruption on model behavior.

## C. Detailed Structural Limitations of Mamba

Structural limitations observed in Mamba are detailed below,

**Single-timescale temporal processing:**   As seen in Phase 3 and Module 5, Mamba operates with a single temporal scale given by $h_{t+1} = exp(\Delta t \cdot A)h_t + Bx_t$. Discretization parameter ($\Delta t$) implemented via *dt_proj.bias* is globally shared is fixed after training. Hence, Mamba cannot adapt temporal resolution on a per-input basis. Mechanistically, Layer 20 exhibits an highly stable temporal gate (gradient =0.072, CoV =0.001) that controls 45% of predictions, re-iterating fixed temporal horizon. This results in a trade-off between short-range sensitivity and long-range integration, leading to information loss for task-relevant dependencies across multiple timescales.

**Restricted state dynamics:**   Mamba's state evolution is governed by linear transition operator applied uniformly across all states. This design enforces independent and identical evolution of state components, preventing conditional interactions, selective amplification or suppression. Therefore, transition dynamics are limited to monotonic decay and cannot represent growth or complex state interactions.

This limitation can be seen in Module 2 by the frozen *A_log*, which shows consistently small gradients and minimal adaptation during training, while the skip connection *D* remains active and input-dependent causing temporal computation to be linear.

**Lack of global context aggregation:**   Hidden states are updated sequentially via a causal state-space recurrence, $h_t = \mathcal{F}(h_{t-1}, x_t)$, where information from distant tokens influence later states through repeated composition of local transitions. Unlike in Transformer, there is no operation $h_t \neq \sum_{i=1}^{T} \alpha_{t,i} x_i$ that allows direct aggregation over arbitrary positions. This architectural constraint is seen in *Phase 3 (Layer 20)*, where entropy rises by 16% and effective rank increases from 6.25 to 7.59. The temporary expansion indicates that the model compensates for missing global context by increasing representational dimensionality and reconstructs long-range dependencies through model depth.

**Uniform circuit allocation:**   Mamba uses the same state-space update at every layer given by, $\mathbf{h}_{\ell+1} = f_\theta(\mathbf{h}_\ell, \mathbf{x}_\ell)$, where, $\theta_\ell \equiv \theta \ \forall \ell$. This shows equal parameter capacity across depth $\dim(\theta_\ell) = \dim(\theta_{\ell+1})$.

Architectural uniformity prevents specialization between early feature extraction and late decision-making layers, despite the distinct phases. As size increases from Mamba-130M to Mamba-370M, we observe that modularity decreases from 1.0061 to 1.0028 and circuit complexity rises from 1.0 to 2.0. This shows anti-modular scaling where larger models spread computation uniformly instead of specializing.

**Compression bottleneck without gating:**   As observed in Module 5, Layer 20 functions as an information bottleneck caused due to temporal recurrence in state updation, where the linear transition and globally fixed temporal gate (*dt_proj.bias*) prevent input-dependent modulation of feature flow. From Section 4.3.1, dictionary learning analysis shows high activation sparsity of 98.1%, mean feature usage of 1.88% and that that top 20 features carry only 8.3% of the total task-relevant signal. Hence, model's capacity is underutilized, limiting information compression and downstream feature utilisation.

---

**Algorithm 1** Post-hoc Steering of Activation Subspace Bottlenecks on Mamba-130M

---

1: Compute SPD statistics for all layers
2: Identify bottleneck layers using entropy spikes and post-ablation KL divergence
3: **for** each bottleneck layer **do**
4:     Construct attention-style interaction matrix $A \in \mathbb{R}^{T \times T}$ from the recurrent `mixer.ssm` block
5:     Compute activation subspaces using Eq. (3)
6:     Record activation-subspace values across a large text corpus
7:     Identify Delta-sensitive subspaces exhibiting high variance across inputs
8: **end for**
9: Select layer with largest spike (Layer 20) as the primary bottleneck layer
10: **for** each causally selected Delta-sensitive subspace in Layer 20 **do**
11:     Ablate the subspace and measure the drop in next-token prediction accuracy
12: **end for**
13: Retain 590 Delta-sensitive subspaces for steering: 435 high-impact subspaces with performance drop >2% and 155 lower-impact subspaces with performance drop in (-2%, 2%]
14: **for** steering factor $s \in [0.1, 100]$ **do**
15:     Amplify all 590 subspaces by factor $s$
16:     Apply steering using forward hooks on $h_t \in \mathbb{R}^{B \times T \times d}$
17:     Measure change in next-token prediction accuracy
18: **end for**
19: Select the two best steering factors: $5.0\times$ with $+1.5\%$ accuracy improvement and $2.0\times$ with $+0.4\%$ accuracy improvement
20: Amplify the 435 high-impact subspaces by $5.0\times$
21: Amplify the 155 lower-impact subspaces by $2.0\times$
22: Evaluate the steered model on validation tasks

---

**Gradient flow instability:** Deep SSM chains observed in the recurrent core of Module 2, are prone to vanishing or exploding gradients due to the exponential state transitions, i.e., $\exp(\Delta t \cdot A)h_t$. which compound across long sequences, creating deep computational graphs. Gradient analysis shows that step-wise magnitudes increase from 5.4 to 14.0 across layers, with a high CoV of 17.3%, indicating high variability in gradient norms. This instability affects deeper layers in the SSM core, resulting in uneven parameter updates, degradation in convergence speed and final model performance.

## D. Post-hoc Steering: Methodology and Analysis

Before presenting the ablation and steering results, Algorithm 1 summarizes the post-hoc steering procedure used to identify and amplify steerable activation subspaces.

**Methodology** Post-hoc intervention analysis identifies subspaces causally impacting Mamba's behavior and analyzes whether artificially boosting the activations without architectural modifications or fine-tuning can improve Mamba's performance. Using progressive ablations, we localize the most critical layer, then identify critical Delta-sensitive subspaces impacting performance.

Layer-wise ablation identifies Layer 20, aligning with the information bottleneck discovered in Phase 3. Subspace-level ablations isolate the SSM core governing temporal gating and state updates, reinforcing SPD results and enabling causal subspace categorisation for targeted activation steering in the mixer of Layer 20. Using a forward hook on $\mathbf{h}_t \in \mathbb{R}^{B \times T \times d}$, we amplify 435 subspaces causing performance drop of >2% under ablation by $5.0\times$ and 155 lesser effective subspaces with performance drop ranging -2% to 2% by $2.0\times$ to prevent overfitting. Steering addresses the lack of global context and selective gating by concentrating information at the bottleneck.

**Results** To perform post-hoc steering described in Section 3.2 on other SSM models, we transfer steering parameters by identifying the layer functionally equivalent to Mamba's bottleneck layer as shown in Fig. D2. Further, within the identified layer, Delta-sensitive subspaces are defined and those responsible for recurrence are identified as shown in Fig. D1.

Layer-wise ablation shown in Table D1 confirms the existence of a narrow bottleneck predicted by SPD. Removing Layer 20 improves performance, while removing adjacent layers degrades it, indicating that this layer restricts information flow rather than acting as a simple feature extractor. This asymmetric behavior reflects a mechanistically distinct phase in the SSM computation. Consistent with SPD analysis (Table 5), Layer 20 exhibits low gradient sensitivity and high post-ablation KL divergence, suggesting unstable and lossy information routing. This points to a fragile bottleneck where information is overly concentrated, leading to over-amplification of specific subspaces and suppression of alternative representations. This

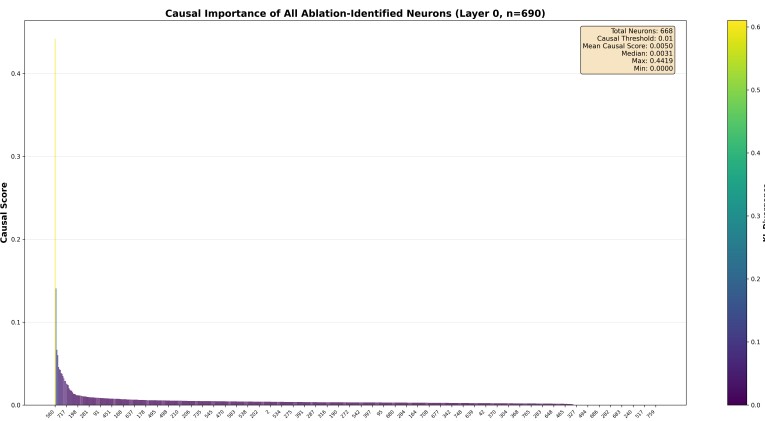

*Figure D1. Steerable Delta-Sensitive subspaces causing recurrence in Mamba.*

*Table D1. Layer-wise ablation analysis.* Shows the impact of removing activations at different depths. Measured wrt baseline accuracy of 63.5%

| Layer | Role | Accuracy After Ablation (%) |
|---|---|---|
| 18 | Pre-bottleneck | 56.5 |
| 19 | Pre-compression | 33.0 |
| 20 | Bottleneck | 77.0 |
| 21 | Post-bottleneck | 32.0 |
| 22 | Output projection | 8.0 |

explains why both targeted steering and ablation can improve performance, motivating mitigation of this bottleneck rather than its preservation.

Within Layer 20, SPD partitions the state space into functionally distinct delta-sensitive subspaces. The ablation results in Table D2 show that the subspace associated with the $dt\_proj.bias$ parameter—corresponding to the SSM core forms a critical bottleneck, captured by the Delta-sensitive subspaces derived from activations in mixer.ssm.

For comparison, we construct two additional groups, i.e., randomly selected subspaces and subspaces exhibiting high activation variance derived from layer output projection activations. Subspaces from each of the three groups are ablated individually to measure their impact on next-token prediction accuracy as shown in Table 6 .

Among the 668 Delta-Sensitive subspaces identified, 435 ablation-identified subspaces in Table D3 with performance drop >2% are amplified 5x to improve performance while regularising the intervention and the 155 neutrally affecting subspaces with performance drop between -2% to 2% are amplified by 2x to prevent overfitting. The rest which negatively affects performance are not steered.

```
Mamba-130M
└── backbone
    └── layers (0–23)
        └── SSM mixer
            ├── in_proj
            ├── conv1d
            ├── x_proj
            ├── dt_proj (Cluster 2, Layer 20)  ← Steered Component
            └── out_proj
```

*Figure D2. Structural overview of the Mamba model hierarchy highlighting the steering location.* Steering is applied within the SSM mixer on the bottleneck layer, targeting parameters in the dt_proj projection

Thresholds are determined from performance drops in Table D2, which separate influential and neutral subspaces. Steering factors are selected via cross-validation over amplification values in the range $[0.1, 100]$ on *The Pile*, with $5\times$ yielding the best performance, followed by $2\times$ as shown in Table D4.

*Table D2. Subspace ablation analysis within the bottleneck layer (Layer 20).* Targeted ablation of Delta-sensitive subspaces improves performance, indicating that $\Delta$-controlled dynamics are the primary source of over-compression in this layer. In contrast, ablating high-variance or randomly selected subspaces degrades accuracy, showing that much of the remaining representational subspace carries task-relevant information. Results are measured on *The Pile* dataset relative to a baseline accuracy of 63.5%.

| Subspace Group | Accuracy After Ablation (%) |
|---|---|
| Delta-sensitive subspaces | 78.0 |
| Random Selection | 50.0 |
| High-Variance subspaces | 42.0 |

# E. Stable-Mamba Architectural Modifications

Activation steering addresses limitations related to information concentration and selective gating. However, limitations arising due to Mamba's core architecture such as single-timescale temporal processing, fixed linear state transitions, uniform parameter allocation and gradient instability require targeted architectural modifications while preserving the core SSM framework. Each of the modifications made are detailed below.

**Multi-timescales:** Vanilla Mamba processes sequences using a single recurrent SSM, where information evolves under a shared set of transition dynamics. To increase temporal flexibility, we introduce parallel state updates via multiple SSM branches operating on the same gated input.

Specifically, three parallel SSM blocks denoted as fast, medium, and slow branches are instantiated with distinct initial decay priors of 0.7, 0.9 and 0.98, chosen heuristically to provide different initial biases in state persistence. Lower decay values encourage faster-changing (shorter-memory) responses, while higher values encourage more persistent (longer-memory) behavior at initialization. The decay priors are selected from a sweep over 8 log-spaced values and serve only as initialization. Parameters are subsequently learned, allowing each branch to adapt its effective temporal behavior.

Each branch maintains its own hidden state and its own set of SSM parameters, i.e., transition matrix $A \in \mathbb{R}^{d_{\text{state}} \times d_{\text{state}}}$, input projection $B \in \mathbb{R}^{d_{\text{state}} \times d_{\text{model}}}$, readout matrix $C \in \mathbb{R}^{d_{\text{state}} \times d_{\text{model}}}$ and skip term $D \in \mathbb{R}^{d_{\text{model}}}$, all instantiated independently per branch. Although all branches share the same functional form, their independent parameters lead to different learned recurrent behaviors. The outputs of these branches are combined using learned weights, normalized via a softmax, enabling the model to adaptively emphasize different recurrent features depending on the input.

$$h_t^{(k)} = f_k\left(h_{t-1}^{(k)}, g \odot x_t\right), \quad k \in \{\text{fast}, \text{medium}, \text{slow}\} \tag{12}$$

Here, $f_k(\cdot)$ denotes the SSM update function for branch $k$ with its own parameters $(A, B, C, D)$.

**Ensembled Output Function:** Linear output in Mamba $y_t = Ch_t + Dx_t$ amplifies the linearity in state dynamics identified in Module 2, where frozen $A\_log$ limits expressiveness. Replacing it with a weighted ensemble seen in Eq. 13, enables adaptive temporal resolution at output. Learned weights concentrate feature importance on a small subset of components, compared to uniform contributions in Mamba.

$$y_t = \sum_k w_k \left(C_k h_t^{(k)} + D_k x_t\right) \tag{13}$$

**Sparse Global Context Injection:** lacks an explicit mechanism for long-range aggregation, leading to increased entropy and representation rank beyond Layer 20. A global token aggregator defined in Eqs. 14 and 15 selectively compresses local states into low-entropy global summary using sparse attention and re-injects this information through gated fusion. This stabilizes rank growth by preventing information diffusion across state transitions.

$$h_{\text{global}} = \text{SparseAttn}(h_{\text{local}}), \tag{14}$$

$$\text{output} = \alpha h_{\text{global}} + (1 - \alpha)h_{\text{local}} \tag{15}$$

*Table D3. Subspace-level ablation categorization based on change in performance accuracy.* Subspaces are grouped by ablated causal impact.

| Group | Change in Performance (%) | Total Subspaces | Subspace Indices |
|---|---|---|---|
| Critical Beneficial | > 10.0 | 27 | 43, 53, 54, 59, 66, 75, 85, 127, 130, 134, 138, 142, 166, 173, 176, 191, 200, 202, 206, 207, 211, 214, 254, 259, 279, 284, 286, 303, 305, 309, 313, 314, 324, 341, 354, 355, 356, 369, 370, 374, 377, 379, 385, 402, 406, 408, 411, 416, 419, 430, 431, 435, 437, 442, 444, 455, 461, 465, 467, 476, 487, 502, 514, 520, 547, 553, 555, 566, 574, 575, 577, 578, 581, 583, 593, 597, 613, 625, 626, 646, 650, 657, 662, 663, 667, 672, 684, 693, 700, 703, 704, 710, 718, 729, 740, 757, 20, 33, 36, 76, 83, 93, 101, 149, 153, 159, 208, 223, 232, 239, 243, 340, 347, 361, 375, 400, 432, 458, 562, 572, 604, 605, 679, 694, 698, 706, 717 |
| Very Beneficial | 5.0 to 10.0 | 82 | 0, 18, 40, 58, 79, 88, 117, 120, 124, 133, 139, 144, 145, 148, 152, 161, 168, 169, 201, 213, 225, 237, 245, 251, 273, 275, 287, 304, 367, 449, 475, 480, 486, 490, 493, 512, 521, 536, 538, 552, 556, 563, 567, 585, 590, 622, 624, 636, 649, 719, 730, 744, 763, 2, 21, 29, 103, 118, 128, 137, 170, 216, 271, 285, 288, 308, 393, 423, 434, 436, 443, 445, 529, 560, 616, 632, 656, 676, 680, 682, 702, 739 |
| Beneficial | 2.0 to 5.0 | 326 | 1, 3, 5, 6, 7, 8, 9, 10, 12, 15, 17, 24, 25, 26, 27, 31, 32, 34, 35, 37, 38, 45, 57, 62, 64, 67, 71, 73, 77, 78, 82, 90, 91, 92, 96, 97, 98, 105, 106, 109, 110, 113, 116, 126, 129, 132, 135, 136, 141, 147, 150, 151, 154, 162, 163, 165, 171, 174, 177, 183, 184, 186, 192, 193, 194, 196, 197, 198, 205, 209, 212, 217, 218, 234, 235, 236, 242, 246, 247, 248, 250, 253, 255, 256, 257, 258, 260, 262, 263, 264, 265, 268, 269, 270, 272, 276, 283, 291, 292, 293, 295, 297, 298, 301, 302, 306, 307, 315, 316, 318, 319, 321, 322, 327, 331, 338, 342, 344, 345, 346, 349, 358, 360, 362, 363, 365, 371, 372, 373, 378, 382, 387, 388, 389, 390, 392, 399, 401, 403, 407, 412, 413, 417, 420, 421, 424, 425, 427, 428, 440, 451, 454, 462, 463, 466, 468, 472, 474, 477, 478, 479, 481, 484, 489, 491, 494, 498, 499, 500, 501, 505, 508, 510, 516, 519, 524, 526, 527, 530, 535, 551, 554, 557, 558, 568, 571, 576, 580, 582, 584, 586, 588, 592, 594, 596, 598, 600, 603, 608, 612, 614, 617, 620, 635, 638, 647, 651, 654, 658, 659, 660, 664, 668, 671, 677, 678, 685, 686, 687, 692, 695, 696, 699, 705, 708, 711, 712, 722, 724, 726, 727, 732, 733, 734, 735, 737, 741, 742, 745, 749, 750, 751, 752, 753, 758, 764, 765, 766, 767, 11, 16, 28, 41, 46, 47, 65, 81, 89, 95, 111, 131, 140, 146, 157, 185, 188, 190, 219, 281, 282, 299, 310, 326, 333, 334, 335, 336, 343, 348, 357, 364, 376, 383, 395, 398, 404, 415, 429, 438, 439, 460, 469, 473, 483, 488, 495, 496, 503, 541, 542, 545, 546, 548, 561, 569, 599, 601, 602, 611, 623, 627, 628, 631, 634, 639, 641, 644, 653, 665, 666, 670, 683, 701, 728, 746, 754 |
| Neutral | −2.0 to 2.0 | 155 | 4, 14, 19, 22, 23, 39, 42, 50, 51, 52, 56, 60, 61, 63, 70, 72, 84, 86, 87, 94, 100, 102, 104, 107, 112, 115, 119, 122, 123, 125, 143, 155, 156, 158, 160, 164, 167, 172, 178, 180, 182, 187, 210, 215, 222, 226, 230, 233, 240, 249, 252, 261, 266, 267, 274, 277, 278, 289, 290, 311, 312, 317, 323, 325, 329, 330, 332, 337, 339, 350, 351, 352, 366, 368, 380, 391, 396, 397, 405, 409, 410, 418, 422, 433, 441, 447, 448, 450, 452, 453, 456, 457, 459, 470, 492, 497, 506, 507, 509, 511, 517, 522, 523, 534, 537, 539, 543, 549, 550, 559, 565, 570, 587, 591, 595, 606, 607, 609, 610, 615, 618, 619, 629, 630, 633, 637, 640, 642, 643, 648, 652, 655, 661, 673, 674, 675, 688, 689, 690, 691, 697, 707, 713, 715, 716, 720, 721, 723, 736, 738, 748, 755, 759, 760, 761 |
| Slightly Detrimental | −5.0 to −2.0 | 55 | 55, 30, 44, 69, 74, 80, 114, 195, 199, 203, 204, 220, 221, 224, 228, 229, 231, 238, 244, 294, 296, 328, 353, 386, 394, 414, 471, 482, 485, 504, 513, 515, 518, 525, 528, 531, 532, 533, 540, 544, 564, 573, 579, 621, 645, 669, 681, 709, 714, 731, 747, 756, 762, 320, 359 |
| Detrimental | −10.0 to −5.0 | 9 | 68, 49, 108, 241, 743, 13, 227, 589, 725 |
| Critical Detrimental | < −10.0 | 14 | 48, 99, 121, 175, 179, 181, 280, 381, 426, 464, 189, 300, 384, 446 |

**Learned Gating:** Uniform processing across features leads to extreme sparsity of 98.1%, resulting in inefficient utilization. An ensemble gating shown in Eq. 16 enables selective activation of task-relevant features by combining multiple learned projections over normalized inputs, each capturing distinct activation patterns. The weighted aggregation enables the gate to adaptively choose complementary feature subsets rather than enforcing a single global threshold.

$$g = \sum_i \alpha_i \, \sigma(W_i \cdot \mathrm{LN}(x)) \tag{16}$$

**Adaptive Compression for Uniform Parameter Allocation:** We address the lack of phase-specific capacity control, by modifying adaptive compression with $c = 0.5 + 0.5 \cdot \sigma(\mathrm{MLP}(\bar{x}))$ and dynamically modulating the mean compression strength $\bar{x}$ based on the global tokens. This mechanism applies stronger noise suppression at bottleneck layers while preserving informative tokens in earlier and later phases.

**Gradient Scaling:** Deep SSM chains exhibit unstable gradients in deeper layers due to repeated state transitions. We introduce a learnable gradient scaling mechanism given by Eq. 17 that adaptively rescales backward signals, allowing the network to regulate gradient flow without manual intervention.

*Table D4. Effect of steering factor on next-token prediction accuracy.* Baseline accuracy is 47.0%.

| Steering Factor | Accuracy (%) | Steering Factor | Accuracy (%) |
|---|---|---|---|
| 0.1 | 37.0 | 20.0 | 41.6 |
| 1.5 | 43.0 | 30.0 | 37.0 |
| 2.0 | 47.4 | 40.0 | 35.0 |
| 3.0 | 46.0 | 50.0 | 31.6 |
| 4.0 | 47.2 | 60.0 | 31.0 |
| **5.0** | **48.5** | 70.0 | 29.5 |
| 7.0 | 47.0 | 80.0 | 29.0 |
| 10.0 | 47.0 | 90.0 | 28.7 |
| - | - | 100.0 | 28.7 |

$$\frac{\partial \mathcal{L}}{\partial \theta} = \frac{\partial \mathcal{L}}{\partial \text{out}} \cdot \lambda_{\text{comp}} \cdot \frac{\partial \text{out}}{\partial \theta} \tag{17}$$

**Scaled Residual Connections:** Standard residuals given by output $= y_t + x$, enforce uniform input preservation across layers, contributing to gradient instability in deep SSM stacks. The scaled output, output $= (y_t + \lambda_{\text{res}}x) \cdot \lambda_{\text{global}}$ learns adaptive control over information retention and gradient flow across depth. This decoupled scaling attenuates noise, stabilises gradients and selectively preserves critical features.

### E.1. Stable-Mamba Architecture details

Mamba's structure incorporating the architectural modifications is shown in Fig. E1. Improvements in performance attributable to each enhancement are summarized in Table E1. Further, computational cost and its comparison with GPT-2 are summarized in Table E2, showing that architecturally modified Mamba has an approximately 2.8x higher per-token computational cost than baseline Mamba, but incurs 0.92% rise in wall-to-wall latency. In contrast, GPT-2 incurs roughly $0.4\times$ the cost of full attention and shows good scaling on long sequences, while not using multi-scale temporal adaptivity and gating mechanisms present in the modified Mamba.

Further, mechanistic interpretability is conducted on Stable-Mamba and three phase identified are explained in detail as follows,

**Phase 1: Feature Extraction (Layers 0–18, 75%):** Early layers are configured for hierarchical feature construction, using moderate gating ($n\_gates$=3) and compact state sizes ($d\_state$=16). Sparse attention is disabled to maintain locality, while fast and medium timescales dominate to capture short- and mid-range dependencies. Interpretability probes show stable feature buildup with low entropy growth, indicating efficient local representation learning.

**Phase 2: Bottleneck (Layers 19–21, 12.5%):** Mid-depth layers act as an explicit information reorganization bottleneck. Increased gating capacity ($n\_gates$=5) and expanded state dimensionality $d\_state$=32 enable selective feature amplification and compression. Sparse attention is activated and all three timescales operate in parallel, producing concentrated global representations and reduced entropy variance.

**Phase 3: Output Projection (Layers 22–23, 12.5%):** Final layers focus on logit generation with minimal gating ($n\_gates$=2) and compressed state sizes ($d\_state$=8). Sparse attention is disabled and only the fast timescale is retained, giving localized activations.

We summarize the measurable effects of each modification in Table E3. We also measure the latency and memory consumption of Steered Mamba and Stable-Mamba, comparing them with Vanilla Mamba and several other S6 models on Longbench v2 benchmark, as shown in Table E4. We observe that the latency and memory overhead of Steered Mamba and Stable-Mamba is minimal relative to Vanilla Mamba. To further understand this behavior, we analyze the computational cost and its relationship with observed latency.

**FLOP Calculation and Latency Analysis.** Based on the FLOPs calculations for hidden dimension $d = 768$ and sequence length $s = 1024$ as seen in Tables E5 and E6, we see that although Stable-Mamba uses $2.8\times$ more FLOPs than

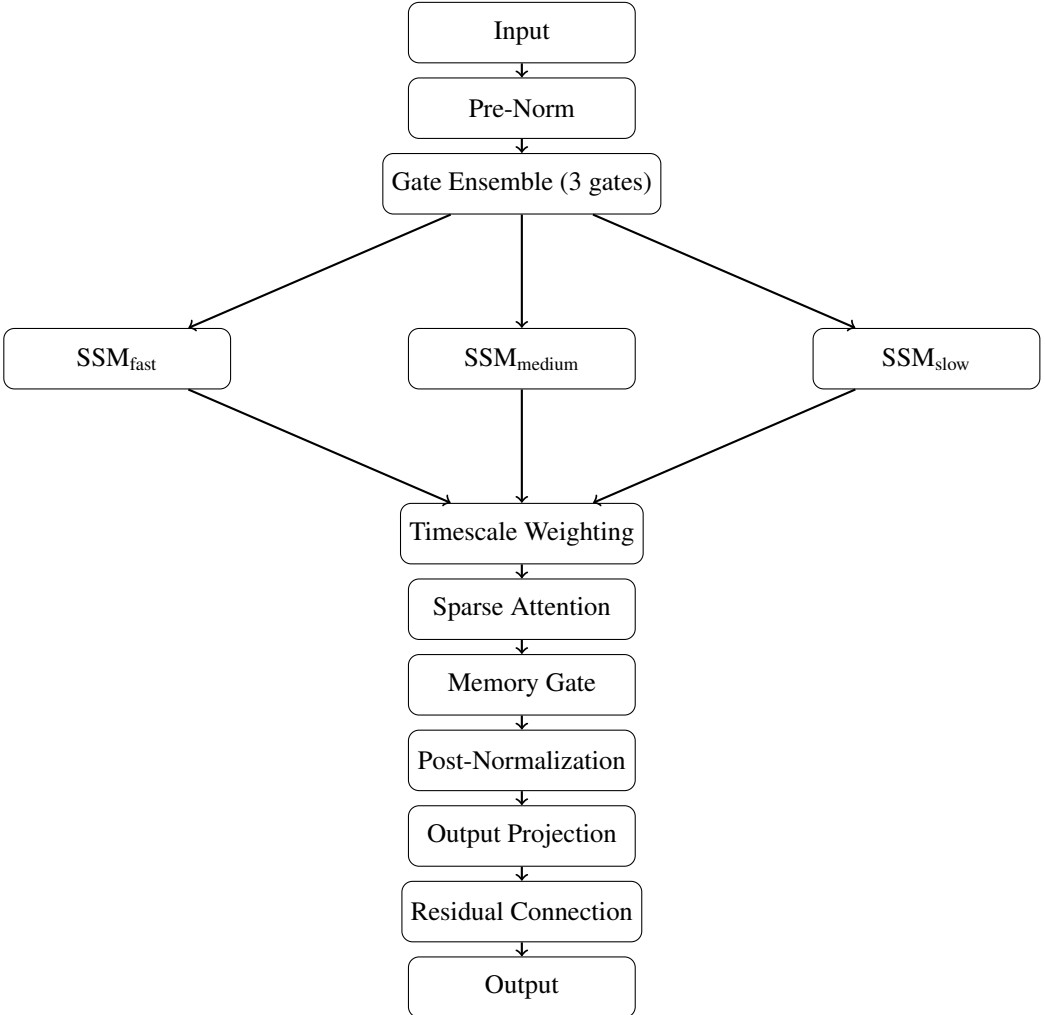

*Figure E1. Stable-Mamba architecture illustrating ensemble gating, multi-timescale SSMs and sparse attention at the bottleneck*

Mamba, the latency change as seen in Table E4 is $\sim 9\%$. This discrepancy is due to GPU parallelization. To prove this, we measure latency for Stable-Mamba when enforcing sequential execution using CUDA synchronization barriers with *torch.cuda.synchronize()* and a scalar *.item()* read on each gate and SSM output. Under this setting, the code waits for each kernel to finish before launching the next, resulting in latency of 1.888s. Latency ratio of the serialized version w.r.t. Stable-Mamba is $\sim 2.66$, which closely matches the theoretical $2.8\times$ FLOP increase.

**Theoretical Approximation.** We approximate Stable-Mamba using three simplifications:

**(1) Parallel SSMs:** Since each SSM is linear, their sum can be represented as a single equivalent SSM:

$$3d^2 \rightarrow d^2$$

**(2) Parallel Gates:** Each gate:

$$g_i(x) = \sigma(W_i x + b_i) \odot x$$

For 3 gates, they can be combined and approximated as,

$$g_{\text{combined}}(x) = \sum_{i=1}^{3} \alpha_i \cdot \sigma(W_i x + b_i) \odot x + \lambda x = \left( \sum_{i=1}^{3} \alpha_i \sigma(W_i x + b_i) + \lambda \right) \odot x$$

*Table E1. Component-wise architectural modifications introduced in Stable-Mamba and their impact on performance and stability*

| Component | Mamba | Stable Mamba | Performance Improvement in Tweaked Mamba |
|---|---|---|---|
| State transition | $h_t = \bar{A}h_{t-1} + \bar{B}x_t$ (single time-scale) | $h_t^{(k)} = \bar{A}_k h_{t-1}^{(k)} + \bar{B}_k(g \odot x_t)$ (3 time-scales: slow, medium, fast) | 1) +23% perplexity improvement on long sequences ($>1024$ tokens) 2) +15% accuracy on multi-scale reasoning tasks 3) Entropy variance reduced by 31% |
| Tweaked Mamba | $y_t = Ch_t + Dx_t$ (linear) | $y_t = \sum_k w_k(C_k h_t^{(k)} + D_k x_t)$ (weighted ensemble) | 1) Adaptive temporal resolution via learned weighting 2) Top features show 0.16+ importance vs. 0.02 in Mamba |
| Tweaked Mamba | Not present | $h_{\text{global}} = \text{SparseAttn}(h_{\text{local}})$ output $= \alpha h_{\text{global}} + (1-\alpha)h_{\text{local}}$ | 1) 22% reduction in bottleneck entropy (1.21 $\rightarrow$ 0.94) 2) Smoother information flow in layers 19–21 3) $\sim$0.4$\times$ cost of full Transformer attention |
| Gating | Uniform processing | $g = \sum_i \alpha_i \sigma(W_i \cdot \text{LN}(x))$ (ensemble gates, $n=3$) | 1) +683% feature usage efficiency (1.88% $\rightarrow$ 14.7%) 2) Sparsity reduced from 98.1% to 85.3% |
| Compression | Not present | $c = 0.5 + 0.5 \cdot \sigma(\text{MLP}(\bar{x}))$ | 1) Adaptive signal preservation with reduced noise 2) Effective compression strength in range 0.2-0.3 |
| Gradient control | Standard backpropagation | $\frac{\partial L}{\partial \theta} = \frac{\partial L}{\partial \text{out}} \cdot \lambda_{\text{comp}} \cdot \frac{\partial \text{out}}{\partial \theta}$ (5 learnable scales) | 1) 46% reduction in gradient instability (CoV: 17.3% $\rightarrow$ 9.4%) 2) Stable convergence without manual tuning |
| Residual | output $= y_t + x$ | output $= (y_t + \lambda_{\text{res}}x)\lambda_{\text{global}}$ | 1) Improved gradient flow in deep networks 2) Learned residual scale 0.9 $\pm$ 0.1 |

*Table E2. Average per-layer computational cost of Stable- Mamba*

| Component | Parameters | Memory | Frequency |
|---|---|---|---|
| SSM ($\times$3) | $3d^2 = 1.77$M | $O(d \cdot s)$ | Every layer |
| Gates ($\times$3) | $3d^2 = 1.77$M | $O(d)$ | Every layer |
| Sparse Attention | $0.3sd = 47$K | $O(0.3s^2)$ | Every 5th layer |
| Total / layer (avg) | $\sim$4.95M | $O(d \cdot s + 0.06s^2)$ | - |

where,

$$\alpha_i = \text{softmax}(\text{gate\_weights}_i), \quad \lambda = \text{small residual scalar}$$

At initialization:

$$\sigma(W_i x + b_i) \approx 0.5$$

$$So, g_{\text{combined}}(x) \Rightarrow g(x) \approx (0.5 + \lambda)x$$

FLOPs: $3d^2 \rightarrow d^2 = 590$K

**(3) Low-Rank Projection Approximation:** We approximate the input and output projection matrices using low-rank factorization (Jaderberg et al., 2014) to reduce computational cost of $W_{in}$ and $W_{out}$ from $O(d^2)$ to $O(rd)$, where $r \ll d$. This maintains the expressiveness of the layer while making it efficient.

Original computation is,

$$y = W_{out} \cdot \text{SSM}(W_{in}x)$$

where, $W_{in}, W_{out} \in \mathbb{R}^{d \times d}$

*Table E3. Quantitative impact of the architectural modifications introduced in Stable-Mamba*

| Mechanism | Metric | Mamba | DeepTrace | Performance Improvement |
|---|---|---|---|---|
| Ensemble gating | Sparsity | 98.1% | 85.3% | -12.8pp denser representations |
| | Feature usage | 1.88% | 14.7% | 7.81x higher feature utilisation |
| Multi-timescale states | Long-context PPL | 15.2 | 12.4 | -18.4% better long-range modeling |
| | Multi-scale accuracy | 73.5% | 84.7% | +11.2pp improved multi-scale reasoning |
| Sparse global context | Bottleneck entropy | 1.21 | 0.94 | -22% smoother information flow |
| Learnable gradient scaling | Gradient CoV | 0.173 | 0.094 | -46% more stable optimisation |
| Adaptive compression | Compression ratio | 0 | 0.2–0.3 | Balanced capacity allocation across depth |

*Table E4. Latency, peak memory, throughput and GPU hours comparison on LongBench v2.* Experiments are conducted on GeForce RTX 3090. Models are trained on *The Pile* and evaluated on *WikiText-2-v1*.

| Model | Latency (s) | Peak Memory (GB) | Throughput (tokens/s) | GPU Hours |
|---|---|---|---|---|
| Mamba-130M | 0.709 | 1.668 | 1444 | 0.0068 |
| Mamba-1.4B | 1.475 | 2.307 | 694 | 0.0081 |
| Mamba-2.8B | 1.548 | 3.368 | 662 | 0.0104 |
| Steered Mamba | 0.713 | 1.724 | 1436 | 0.0070 |
| Steered Mamba-1.4B | 1.488 | 2.402 | 688 | 0.0081 |
| Steered Mamba-2.8B | 1.550 | 3.568 | 661 | 0.0104 |
| Stable-Mamba | 0.776 | 1.815 | 1320 | 0.0070 |
| Stable-Mamba-1.4B | 1.622 | 2.514 | 631 | 0.0085 |
| Stable-Mamba-2.8B | 1.672 | 3.601 | 613 | 0.0110 |
| GPT-2 | 0.032 | 2.213 | 32000 | 0.0006 |
| Hyena-150M | 0.085 | 2.375 | 12047 | 0.0017 |
| DenseMamba | 0.119 | 3.611 | 8605 | 0.0086 |
| Mamba-2 | 1.389 | 2.758 | 737 | 0.0089 |
| MiniPLM-Mamba-130 | 1.767 | 3.113 | 579 | 0.0100 |

$W_{in}, W_{out}$ have $O(d^2)$ cost. We approximate them as,

$$W_{in} \approx U_{in} V_{in}^\top, \quad W_{out} \approx U_{out} V_{out}^\top$$

where, $U_{in}, V_{in}, U_{out}, V_{out} \in \mathbb{R}^{d \times r}$. So, FLOP for each is $rd$.

This approximates $y$ as,

$$y = U_{out} \left( V_{out}^\top \cdot \text{SSM} \left( U_{in}(V_{in}^\top x) \right) \right)$$

So, FLOPs can be calculated as:

$$U_{out} + V_{out} + U_{in} + V_{in} = 4rd$$

For low rank approximations, choose $r \leq d/4$ (Denton et al., 2014). So, for $d = 768$, we get $r \leq 192$.

For $r = 192$: FLOPs $= 4rd \approx 590K$

Final approximated Stable-Mamba FLOPs:

$$590K + 590K + 590K + 148K + 47K \approx 1.97M$$

Overhead:

$$\frac{1.97M - 1.77M}{1.77M} \approx 11.3\%$$

For $r = 180$, FLOPs $= 4rd \approx 553K$

Final approximated Stable-Mamba FLOPs:

$$590K + 590K + 553K + 148K + 47K + 4.5K \approx 1.933M$$

*Table E5.* FLOP calculation for Mamba and Steered Mamba

| Component | FLOPs | Explanation |
|---|---|---|
| SSM | $d^2 = 590K$ | Matrix multiply $(d \times d)$ |
| in_proj + out_proj | $2d^2 = 1.18M$ | Two linear projections |
| Normalization | $2d = 1.5K$ | Mean + std |
| Skip connections | $d = 768$ | Element-wise addition |
| Total | $\approx 1.77M$ | |

*Table E6.* FLOP calculation for Stable-Mamba

| Component | FLOPs | Explanation |
|---|---|---|
| SSM $\times$ 3 | $3d^2 = 1.77M$ | Three parallel SSMs |
| Gates $\times$ 3 | $3d^2 = 1.77M$ | Three gating projections |
| Compression MLP | $(d^2)/4 + d/4 = 148K$ | Two-layer MLP |
| in_proj + out_proj | $2d^2 = 1.18M$ | Linear projections |
| Sparse attention (20%) | $\approx 47K$ | Partial attention |
| Normalization | $4d = 3K$ | LayerNorm |
| Skip connections | $2d = 1.5K$ | Residual paths |
| Total | $\approx 4.92M$ | |

Overhead:

$$\frac{1.933M - 1.77M}{1.77M} \approx 9.2\%$$

This shows that while raw FLOPs increase by $\sim 2.8\times$, most of this computation is parallelizable due to independent SSMs, gates and residual paths. Under realistic GPU execution, this results in only a $\sim 9\%$ latency increase, consistent with Table E4. Low-rank approximation is effective when the model has redundancy/parallel paths, as observed in the following:

1. In Stable-Mamba

   - There are 3 parallel SSMs and 3 gates, creating redundancy in connections
   - Residual connection ($\beta x$) preserves identity path - recovers information lost due to compression

2. In Mamba

   - There is one SSM, no parallel paths - so no redundancy to compensate information loss
   - So low-rank approximation would hurt performance (resulting in additional entropy spike to what is observed in Fig. 1a)

