# OpenReview forum: "Interpreting and Steering State-Space Models via Activation Subspace Bottlenecks"
_ICML.cc/2026/Conference — ICML 2026 regular_

### Official Review · Reviewer_pojf · 2026-03-09

**Soundness:** 2
**Presentation:** 2
**Significance:** 3
**Originality:** 3
**Overall Recommendation:** 5
**Confidence:** 3

**Summary:**

This paper proposes an interpretability pipeline for Mamba-family SSMs centered on identifying Activation Subspace Bottlenecks, with two downstream applications:
1. Using multiple interpretability techniques (SAE, APD, SPD) to identify the existence of activation space bottleneck: a small number of activation subspace in mamba that are responsible for routing information. The key finding is a sharp entropy spike at Layer 20, driven by a near-frozen dt_proj.bias parameter that acts as a master temporal gate.
2. A post-hoc steering method inspired by this discovery to remove this bottleneck
3. Stable-Mamba, a set of minimal architectural modifications (adding only 256 parameters) that address deeper structural limitations not fully resolved by steering. Stable-Mamba is retrained from scratch and serves as validation that the identified bottlenecks are indeed hindering performance.

The main finding is the activation bottleneck subspace, which is then used to perform steering (novelty 2) and stable mamba (training from scratch, novelty 3)

**Compliance With Llm Reviewing Policy:**

Affirmed.

**Final Justification:**

check acknowledgement comment

**Key Questions For Authors:**

- Check weakness above
- Line 262-263 column 2: paragraph title is duplicated.
- Consider including a Mamba primer before 3.1.2? thus H and T can also be related to the architecture definition.
- Some key notations is not defined: $H$ and $T$ in section 3.1.2 is H the number of heads, and T the sequence length?
- How sensitive are the results to the choice of 8 clusters? Have you tried other values?
- A lot of sections feel text-heavy.
    1. The interp methods used (Section 3.1.2): might be more clear to be presented as a table of Method | method summary | key findings
    2. Section 3.2 can also be summarized by an Algorithm Table

**Limitations:**

Yes, discussed after the conclusion section

**Strengths And Weaknesses:**

Strengths:
- Important finding with convincing validation (simple intervention, Table 1 perplexity changes)
- Interpretability result that also proposes an actionable insight
- Clear contributions from previous works

Weaknesses:
- Missing computational cost study. While I understand that the interpretability techniques run (main compute cost) is 1-time per model, It will be nice to know for people in the community that want to use the proposed method to analyze other models. The SAE training setup and dictionary learning details are specified, but wall-clock time or GPU-hours for the full pipeline are absent.
- Acknowledged in the limitation: but the StableMamba method is only applied to a single model of 130M params (I understand steering has 5 models, but as also discussed in the paper, the steering method is not the strongest actionable from this paper). Results can be made stronger by adding 1-2 more models.
- The steering hyperparameters (which subspaces, what scalar) are tuned on The Pile, then evaluated on benchmarks. But the 5×/2× factors and the 435/155 subspace split feel somewhat arbitrary. The paper doesn't discuss sensitivity to these choices beyond a brief grid search mention. Similarly, the eight-cluster KMeans decomposition lacks justification for why eight clusters specifically.
- On evaluation: The benchmarks used for steering (SQuAD, TriviaQA, DROP, etc.) are fairly standard NLP tasks, but the paper claims gains in "long-context" performance. The long-context benchmarks (RULER, Long Range Arena, LongBench v2) are only used for Stable-Mamba in Table 7. There's a slight disconnect between the claimed motivation (long-context limitations of SSMs) and where steering is actually evaluated.
- On stable mamba:  the paper's narrative read like "we found a bottleneck via interpretability, then we modified the architecture to address it, and performance improved: therefore the bottleneck was real and limiting." But Stable-Mamba doesn't just address the bottleneck. It adds multi-timescale states, sparse global context attention, ensemble gating, adaptive compression, gradient scaling, and scaled residuals all at once. Many of these are generally useful architectural improvements that would likely help any model regardless of whether a bottleneck exists.

Final notes: The core finding is well-supported and the steering intervention is compelling in its simplicity and generalizability. However, the Stable-Mamba contribution bundles too many changes to cleanly validate the bottleneck hypothesis, and the full pipeline is only demonstrated at 130M scale. I would not argue strongly against a weak reject from another reviewer who finds the single-scale evaluation insufficient to support the paper's broader claims.

---

> ### Author Rebuttal · Authors · 2026-03-31
>
> Re: Computational Cost (W1): We thank the reviewer for the comment. We note that Table F4 reports wall-clock time and memory for next-token prediction on IFEval. We have included GPU hours below which we will update in Table F4. Tested on GeForce RTX 3090. Models are trained on The Pile and predicted on Wikitext-v2.
>
> Variant               | GPU hours
>
> Mamba              | 0.0068
>
> Steered Mamba | 0.0070
>
> Stable-Mamba   | 0.0070
>
> DenseMamba    | 0.0086
>
> Mamba-2           | 0.0089
>
> MiniPLM            | 0.0100
>
> Hyena-150M     | 0.0017
>
> Re: Stable architecture scaling (W2): Answered in Reviewer 2’s response - Re: Scaling of steering (W1)
>
> Re: Choice of Hyperparameter and K (W3,Q5): Thank you for the comment. We clarify that the selection of subspaces (435/155) and steering factors (5×/2×) is based on ablation impact and cross-validated scaling.
> - Subspaces are grouped according to ablation-induced performance change (Table E3). This shows that 435 subspaces cause a performance drop greater than 2%, while 155 subspaces lie within the neutral range (−2% to 2%). We use this distribution to distinguish strong and weak causal subspaces
> - Steering factors are selected via a grid search over amplification values (0.1–100) on ‘The Pile’ (Section 3.2). We find that stronger amplification (5×) is optimal for highly causal subspaces (>2% drop) with next token prediction accuracy improvement of 1.5%, while milder amplification (2×) is sufficient for neutral subspaces, preventing over-amplification and instability and ablation performance improvement of 0.4%. Table containing next token prediction accuracy is given below. We will revise the paper to make this connection to Table E3 and include the steering table. Table present in Reviewer 3’s response Re: Steering factor not defined (W7, Q8)
>
> As noted above, k = 8 in KNN was selected as a balance between granularity and robustness. We find that nearby values produce similar qualitative patterns in subspace grouping and steering effectiveness, suggesting the results are not highly sensitive to this choice. We will clarify this in the revision.
>
> Re: Long context claim (W4): We clarify that steering is evaluated on long-context tasks in two places.
> - In Fig. 3, steering is evaluated on RULER, which is a long-context benchmark
> - Table 7 includes Steered Mamba alongside Vanilla and Stable-Mamba, evaluated on long-context benchmarks (RULER, Long Range Arena, LongBench v2). This shows that the gains from steering extend beyond standard NLP tasks to long-context settings as well.
>
> We will revise the paper to highlight that steering improves long-context performance and that these gains are further validated across multiple long-context benchmarks in Table 7.
>
> Re: Stable-Mamba uniqueness (W5): Thank you for the comment. We agree that Stable-Mamba introduces multiple architectural changes. Hence improvements cannot be attributed solely to bottleneck mitigation.Our intended evidence chain is as follows:
> - Fig 1a identifies a clear entropy bottleneck in Vanilla Mamba
> - Post-hoc steering directly targets this bottleneck and reduces it in Fig 1b without any architectural modification. This intervention improves performance, providing the primary evidence that the bottleneck is real and limiting
> - Stable-Mamba is presented as a complementary, architecture-level follow-up. It incorporates modifications (Table F1, Appendix D) motivated by the same failure modes identified through interpretability (Appendix C). It also results in a reduction of the entropy spike (Fig 1c) along with improved performance after retraining (Table 7)
>
> Since these modifications are applied jointly and due to computational constraints we conducted a single training run that bundles several potentially beneficial improvements. We do not claim to isolate the contribution of any single component. Instead, we present Stable-Mamba as a holistic design guided by interpretability insights (Appendix C), demonstrating that architecture-level interventions can effectively address the identified failure modes
>
> Re: Address highlighted weaknesses (Q1): Please see the above responses for each of the weaknesses highlighted.
>
> Re: Repeated title (Q2): We thank the reviewer for the observation and will correct the repeated title.
>
> Re: Mamba primer and notation definition (Q3, Q4): We will add a detailed primer in the appendix and a shorter version in the main text, including Mamba notation regarding sequence length (T), number of heads (H) used in the attention-mapping approximation and​ as the hidden state (h_t) at time step (t)
>
> Re: Presentation improvements (tables, algorithm, readability) (Q6):
> We agree and will improve clarity by:
> - Adding a table summarizing interpretability methods (method | description | key findings)
> - Including an algorithm-style summary for the steering procedure (Section 3.2)
> - Reducing text-heavy sections and improving structure for readability

---

> > ### Author Rebuttal · Reviewer_pojf · 2026-04-02
> >
> > thank you for the rebuttals. most of the concerns have been resolved, except for the scale. I understand on the compute limitations, but only having only 1 model in any paper always raise a concern on my end. I will raise my score, for the novelty of the work (mechanistic insight for mamba); but my concern on a single scale model remains

---

> > > ### Author Response · Authors · 2026-04-08
> > >
> > > We thank the author for the comment. We have added results for steering and stable versions of Mamba-1.4B and Mamba-2.8B.
> > >
> > > Fig 3 (Wo: Without Steering, W: With Steering)
> > > | Model       | Simple Recall (%) (Wo/W) | Instruction Following (%) (Wo/W) | Long Context Recall (%) (Wo/W) | Answering Query (%) (Wo/W) | Chain Reasoning (%) (Wo/W) | Basic Reasoning (%) (Wo/W) |
> > > |-------------|----------------------|------------------------------|----------------------------|------------------------|------------------------|------------------------|
> > > | Mamba-1.4B  | 99.0 / 99.0         | 77.0 / 80.5                 | 40.5 / 45.0               | 35.0 / 45.0           | 22.5 / 26.0           | 28.5 / 32.0           |
> > > | Mamba-2.8B  | 98.0 / 100.0        | 58.0 / 65.5                 | 68.0 / 76.0               | 35.0 / 45.0           | 45.0 / 52.0           | 31.0 / 43.0           |
> > >
> > > Table 7
> > >
> > > LONGBENCH
> > > | Model                  | Accuracy (%) | Confidence (%) | Calibration (%) | Faithfulness (%) | Robustness (%) |
> > > |------------------------|----------|------------|-------------|--------------|------------|
> > > | Mamba-1.4B             | 100.0    | 24.22      | 88.95       | 100.0          | 100.0      |
> > > | Mamba-2.8B             | 100.0    | 26.55      | 95.63       | 100.0          | 100.0      |
> > > | Steered Mamba-1.4B     | 100.0    | 38.33      | 91.00       | 100.0          | 100.0      |
> > > | Steered Mamba-2.8B     | 100.0    | 40.42      | 95.02       | 100.0          | 100.0      |
> > > | Stable-Mamba-1.4B      | 100.0    | 42.14      | 91.17       | 100.0          | 100.0      |
> > > | Stable-Mamba-2.8B      | 100.0    | 45.32      | 96.11       | 100.0          | 100.0      |
> > >
> > > LRA
> > > | Model                  | Image (%) | ListOps (%) | Pathfinder (%) | Retrieval (%) |
> > > |------------------------|-------|---------|------------|-----------|
> > > | Mamba-1.4B             | 26.5  | 65.0    | 65.0       | 68.0      |
> > > | Mamba-2.8B             | 49.5  | 74.0    | 70.0       | 69.0      |
> > > | Steered Mamba-1.4B     | 38.2  | 78.0    | 70.0       | 68.1      |
> > > | Steered Mamba-2.8B     | 50.5  | 81.0    | 70.0       | 68.5      |
> > > | Stable-Mamba-1.4B      | 42.6  | 83.0    | 77.0       | 70.0      |
> > > | Stable-Mamba-2.8B      | 55.5  | 85.5    | 100.0      | 70.0      |
> > >
> > > RULER
> > > | Model                  | NIAH (%)  | Aggregation (%) | QA (%)    |
> > > |------------------------|-------|-------------|-------|
> > > | Mamba-1.4B             | 87.71 | 71.00       | 94.00 |
> > > | Mamba-2.8B             | 90.43 | 70.00       | 95.00 |
> > > | Steered Mamba-1.4B     | 91.00 | 72.00       | 95.50 |
> > > | Steered Mamba-2.8B     | 94.50 | 70.00       | 96.00 |
> > > | Stable-Mamba-1.4B      | 91.55 | 100.00      | 100.00 |
> > > | Stable-Mamba-2.8B      | 93.75 | 100.00      | 99.00 |
> > >
> > > Table F4
> > > | Model                  | Latency (s) | Peak Memory (GB) | GPU Hours |
> > > |------------------------|-------------|------------------|-----------|
> > > | Mamba-1.4B             | 1.475       | 2.307            | 0.0081    |
> > > | Mamba-2.8B             | 1.548       | 3.368            | 0.0104    |
> > > | Steered Mamba-1.4B     | 1.488       | 2.402            | 0.0081    |
> > > | Steered Mamba-2.8B     | 1.550       | 3.568            | 0.0104    |
> > > | Stable-Mamba-1.4B      | 1.622       | 2.514            | 0.0085    |
> > > | Stable-Mamba-2.8B      | 1.672       | 3.601            | 0.0110    |

---

### Official Review · Reviewer_xTr2 · 2026-03-09

**Soundness:** 3
**Presentation:** 2
**Significance:** 2
**Originality:** 2
**Overall Recommendation:** 4
**Confidence:** 3

**Summary:**

This paper presents an interpretability study of Mamba-style state space models (SSMs) to investigate why they still underperform on tasks such as in-context learning and long-context retrieval. To analyze token interactions, the authors apply an attention-mapping procedure that reconstructs token-to-token influence matrices from the SSM dynamics. Using these influence patterns, they identify activation subspaces and demonstrate the presence of activation subspace bottlenecks that restrict information flow.

The identified subspaces are then used as steering targets to modify the model’s behavior. The paper also proposes a set of architectural improvements to Mamba models, referred to as Stable-Mamba. Across five SSM architectures and six benchmarks, the authors show that both the steering approach and the proposed architecture lead to consistent performance improvements, with an average gain of 8.27%.

**Compliance With Llm Reviewing Policy:**

Affirmed.

**Final Justification:**

The paper makes a genuine contribution to SSM interpretability with a lightweight, training-free steering intervention supported by converging evidence across multiple tools. The rebuttal adequately addressed my main concerns around metric definitions, SAE design choices, and steering factor justification.
My remaining concern, shared across reviewers, is the single-scale evaluation and the inability to isolate Stable-Mamba's individual contributions, both acknowledged by the authors as compute-limited. These limit confidence in the broader claims but are reasonable scoping decisions for a mechanistic study.
I maintain 4 (Weak Accept). The core findings are solid and the rebuttal was thorough, but the evaluation scope prevents a higher score.

**Key Questions For Authors:**

- The paper reports several metrics (e.g., entropy, sensitivity, correlation), but their exact definitions are not provided. Could the authors clarify how these metrics are computed?
- Table 3 reports correlation values between several quantities. Could the authors clarify how these correlations are computed (e.g., Pearson/Spearman) and over which variables or samples?
- Line 167 defines an activation subspace in a way that appears closer to a vector than a subspace. Could the authors clarify the exact definition of an activation subspace in this work?
- The introduction suggests that the method identifies a small number of meaningful activation subspaces, while the experiments appear to identify hundreds. Could the authors clarify how these two statements relate?
- The paper applies dictionary learning on top of the SAE representations. Could the authors clarify the motivation for this step and what additional structure it provides beyond the SAE features?
- The SAE used in the experiments appears to use a compressive bottleneck (expansion factor <1). Could the authors clarify the motivation for using a compressive SAE rather than the expanding SAEs typically used in interpretability work?
- The paper selects high-variance subspaces as delta-sensitive subspaces. Could the authors elaborate on the motivation for this choice?
- In Step 4 (Line 196), subspaces are amplified according to a heuristic rule based on performance drop (>2%). Could the authors clarify how this threshold was chosen?
- The paper states that dt_proj.bias is known to force information into a small number of recurrent states (Line 182). Could the authors provide references or further explanation supporting this claim?
- The connection between the interpretability analysis and the proposed Stable-Mamba architecture is not entirely clear. Could the authors elaborate on how the identified bottlenecks directly motivated the architectural modifications?
- There appears to be a repeated paragraph title in Line 262.

**Limitations:**

yes

**Strengths And Weaknesses:**

# Strengths:
- One of the first works on interpretability of SSMs.
- Both steering and architectural modifications lead to measurable performance improvements.
- The methods were evaluated on a fair number of benchmarks and models
- The modifications are in general surgical and only involve a small subspace of the model activation subspace or a small number of parameters/

# Weaknesses:
- Several metrics used throughout the paper (e.g., entropy, sensitivity, correlation) are not clearly defined, making it difficult to interpret the reported results.
- The definition of “activation subspace” (Line 167) appears to correspond to a vector rather than a subspace, which is conceptually confusing.
- The proposed Stable-Mamba architectural modifications appear weakly connected to the interpretability analysis. The interpretability and architecture sections read largely as independent contributions.
- The use of dictionary learning on top of SAEs is not well motivated or explained.
- The SAE used in the experiments appears to use a compressive bottleneck (expansion factor <1), which differs from standard sparse autoencoder setups used in interpretability. This design choice is not justified.
- In line 182, it was mentioned that "dt proj.bias is known to force information into a few recurrent states" but no reference was given.
- In step 4 (line 196), the rule: "Delta-sensitive subspaces with performance drop of >2% and ±2% selected via ablation are amplified 5× and 2× respectively" is also not justified.
- The paper selects high-variance subspaces as delta-sensitive subspaces, but the rationale behind this choice is unclear. What happens if low-variance subspaces are used instead?
- The correlations in table 3 are not defined and it's hard to follow what they mean.
- Results are on limited small sizes and it's not clear if the patterns generalize to larger models.

---

> ### Author Rebuttal · Authors · 2026-03-31
>
> Re: Metric definitions (W1, Q1):
> We have clarified the definition of all metrics in the paper as given below
> -Entropy:SPD-based normalized attribution distribution per layer
> -Sensitivity:Output change vs input perturbation
> -Correlation:Pearson correlation with interpretable properties
> -Universality:U=μ/(1+σ^2)
> -Causal Effect:KL divergence under ablation
> -Delta-sensitivity:Variance of Δ-conditioned activations
>
> Re: Definition of Activation Subspace (W2,Q3):
> Eq 3 produces a weighted aggregation of hidden states,yielding a vector in representation space
> However, subspace refers specifically to low-dimensional feature directions learned via SAE/dictionary learning that capture structured components of activation space.Aggregated vector can be interpreted as a projection onto these directions
> We will revise terminology
> -Feature subspaces:learned via SAE/dictionary learning
> -Aggregated activation:analysis/steering
>
> Re: Stable-Mamba (W3,Q10):
> Architectural modifications in Stable-Mamba are motivated by interpretability analysis(Appendix C).Failure modes in Mamba
> -Compression bottleneck, Layer 20:high entropy, KL divergence (Table 5, Fig 1)
> -extreme sparsity and low feature utilization:Table 4
> -unstable gradient and rigid parameter:Tables 5,B5
>
> Each modification (Table E1) addresses them improving entropy,usage and stability.We will revise to make this mapping between interpretability findings
>
> Re: Dictionary learning motivation (W4,Q5):
> SAEs extract latent features(Table 2) and quantify importance for model behavior, but do not capture how they are used in representation
> Citation: Seonglae Cho et al. 2025. FaithfulSAE: Towards Capturing Faithful Features with Sparse Autoencoders without External Datasets Dependency
> Dictionary learning complements this by decomposing SAE latents into interpretable components and quantifying feature usage (Table 3), enabling assessing if important features are effectively used.It is further complemented by SPD (Section 3.2) by localizing bottlenecks (Fig 1. Layer 20,Table 5)
> Together, SAE captures feature importance.Dictionary learning captures utilization and efficiency.SPD identifies bottlenecks.We will revise paper to clarify complementary role
>
> Re: Compressive bottleneck (W5):
> Our SAE uses a compressive bottleneck,unlike prior interpretability setups.Goal is not to learn a large monosemantic dictionary, but to obtain compact, low-dimensional summary of activations capturing dominant variation
> Citation: Goodfellow et al. (2016). Deep learning. MIT Press. http://www.deeplearningbook.org
> Sparsity promotes selective feature activation, identifying salient subspaces influencing behavior (Table 2).SAE acts as a stable low-rank projection, while dictionary learning operates on them, recovers interpretable components and analyze feature utilization (Table 3).We will revise to clarify design choice
>
> Re: dt_proj.bias reference (W6, Q9):
> dt_proj.bias governs recurrent state updates:∆=τ_∆(Parameter+s_∆(x)),where Parameter refers to bias (Appendix B.7.2).We will revise and add citation.
> Citation: Gu et al. Mamba: Linear-time sequence modeling with selective state spaces. 2024. URL https://arxiv.org/abs/2312.00752
>
> Re: Steering factor not defined (W7,Q8):
> Thresholds and steering factors: empirically determined(Section 4.2.2,Appendix E)
> Delta subspaces are identified via SPD and clustering.Importance evaluated by drop in next-token accuracy via ablation.Gives clear separation:subspaces with >2% drop and within ±2%(Table E3).So thresholds and steering factors are data-driven
> Steering factors (5×, 2×) with max performance are selected by cross-validation over amplification values (0.1–100) on The Pile.We will include this table in the revision
>
> baseline:47.0%
>
> Steering factor| Next token accuracy(%)
>
> 0.1| 37.0
>
> 1.5| 43.0
>
> 2| 47.4
>
> 3| 46.0
>
> 4| 47.2
>
> 5| 48.5
>
> 7| 47.0
>
> 10| 47.0
>
> 20| 41.6
>
> 30| 37.0
>
> 40| 35.0
>
> 50| 31.6
>
> 60| 31.0
>
> 70| 29.5
>
> 80| 29.0
>
> 90| 28.7
>
> 100| 28.7
>
> Re: Delta subspace selection (W8,Q7):
> Variance:Responsiveness to Δ-dependent updates.High-variance subspaces with strong input fluctuations influence output.Low-variance ones with less input variation contribute minimally.Additional results are given in Table 6(Reviewer 2’s response - Re: Steering baseline (W4,Q3))
>
> Re: Table 3 clarification (W9,Q2):
> As discussed in W4, Table 2 shows correlation-based SAE feature importance.Table 3 shows dictionary learning metrics showing feature usage and representation efficiency
> Table 2 focuses on importance,whereas Table 3 captures how effectively SAE features are used in representation.We will revise captions of Tables 2, 3 and text to clarify this distinction
>
> Re: Result scaling (W10):
> Answered in Reviewer 2’s response -Re: Scaling of steering(W1)
>
> Re: Few subspaces claim (Q4):
> Many subspaces are identified initially.Only a subset shows strong causal influence via ablation(Table E1).We will clarify in Introduction
>
> Re: Repeated title(Q11):
> Sorry about this, we have corrected the repeated title

---

> > ### Author Rebuttal · Reviewer_xTr2 · 2026-04-03
> >
> > Thank you for the detailed responses. The additional analyses and clarifications have strengthened the paper. However, I will maintain my current score, as it still reflects my overall assessment.

---

### Official Review · Reviewer_Zb5o · 2026-03-11

**Soundness:** 3
**Presentation:** 3
**Significance:** 3
**Originality:** 3
**Overall Recommendation:** 4
**Confidence:** 4

**Summary:**

This paper investigates the internal computation of Mamba-family state-space models (SSMs) through mechanistic interpretability methods, identifying "Activation Subspace Bottlenecks"—specific layers and parameter clusters where information is forced through a narrow set of activation subspaces. Using Sparse Autoencoders (SAEs), Attribution-based Parameter Decomposition (APD), and Stochastic Parameter Decomposition (SPD), the authors identify Layer 20 of Mamba-130M as a critical bottleneck characterized by an entropy spike and extremely high post-ablation KL divergence.

The authors propose two interventions:

1. **Post-hoc Steering (Section 3.2)**: A test-time intervention that amplifies delta-sensitive subspaces by scalar factors (2x or 5x). This achieves an average 8.27% improvement across 5 SSM architectures and 6 benchmarks without task-specific tuning.
2. **Stable-Mamba (Section 3.3)**: A modified architecture (adding ~256 parameters) featuring multi-timescale dynamics, sparse attention, and learned gating. Retrained from scratch, it yields further gains on long-context benchmarks.

**Compliance With Llm Reviewing Policy:**

Affirmed.

**Key Questions For Authors:**

- Does steering remain effective at larger model scales (e.g., Mamba-370M, 790M, or 1.4B)? Why it matters: All steering results are at 130M. If the bottleneck is an artifact of small capacity, the method's relevance to frontier SSMs is limited.

- How do you reconcile the fact that ablating Layer 20 improves performance (Table E1) with the claim that it is a critical information routing bottleneck? Why it matters: If removing the layer helps, it suggests the layer is detrimental. Clarifying whether this is a "bottleneck" (constraint) or a "bug" (malfunctioning layer) is critical for the mechanistic narrative.

- How does the steering method compare to simple baselines, such as uniformly amplifying all subspaces at Layer 20? Why it matters: This would confirm whether the fine-grained selection of delta-sensitive subspaces is necessary, or if simply boosting signal at the bottleneck layer is sufficient.

**Limitations:**

Yes

**Strengths And Weaknesses:**

# Strengths

## Soundness

- Converging Evidence for Bottlenecks: The identification of the Layer 20 bottleneck is robust, supported by multiple independent metrics: SPD entropy spikes (Fig. 1a), KL divergence of 813.0 vs. ~1.0 for neighbors (Table 5), and SAE sparsity statistics.
- Principled Steering Selection: Ablation studies (Table 6) convincingly validate the steering target. Steering the identified cluster (Cluster 2) at Layer 20 preserves/improves performance, whereas steering the same cluster at other layers or random clusters at Layer 20 degrades it.
- Fragility of Knowledge: The finding that perturbing delta-sensitive subspaces causes up to +474% perplexity increase in Mamba (vs. minimal change in Transformers) provides striking evidence that SSM knowledge encoding is far more concentrated and fragile (Table 1).

## Significance

- Practical Utility of Interpretability: The paper successfully bridges mechanistic interpretability with practical model improvement. The post-hoc steering method is lightweight, requires no training, and generalizes across 5 SSM architectures (Fig. 3), making it practically deployable.
- Architectural Insights: The identification of dt_proj.bias as a "frozen master gate" (Appendix B.7.2) that locks in early during training is a specific, actionable insight for future SSM architecture design.

## Presentation

- Comprehensive Appendix: The 22-page appendix serves as an excellent reference for Mamba internals, covering universality, causal subspaces, induction heads, and scaling analysis.
- Clear Narrative: The five-phase decomposition of Mamba's computation (Appendix C) offers a coherent narrative of information flow.

# Weaknesses

## Soundness

- Limited Model Scale: All primary evaluations are performed on 130M-parameter models. While scaling analysis (Appendix B.6) suggests the bottleneck location is scale-invariant (~85% depth), the effectiveness of steering is not tested on larger, more capable models (e.g., Mamba-1.4B, 3B, or 7B).
- Ambiguous Causal Interpretation of Layer 20: Table E1 shows that ablating Layer 20 improves accuracy to 77.0% (from 63.5% baseline). This suggests the layer may be actively harmful rather than just a "bottleneck" or choke point. The paper does not fully reconcile why a "critical routing" layer improves performance when removed.
- Confounded Stable-Mamba Evaluation: Stable-Mamba introduces six simultaneous modifications (multi-timescale dynamics, sparse attention, learned gating, etc.). It is difficult to attribute gains specifically to interpretability-derived insights versus generic architectural improvements like sparse attention.
- Lack of Steering Baselines: The evaluation lacks simpler baselines, such as uniform amplification of all subspaces at Layer 20 or random subspace amplification. This makes it hard to isolate the value of the specific delta-sensitive subspace selection.

## Significance

- Modest Gains on Non-Mamba Models: While Mamba-130M sees a 20.13% boost, improvements on other architectures (Hyena, Mamba-2) are often small or negligible (Fig. 3). The "average 8.27%" figure is heavily skewed by the base Mamba result.

---

> ### Author Rebuttal · Authors · 2026-03-31
>
> Re: Scaling of steering (W1): Thank you for the insightful comment. We agree that evaluation at larger scales would strengthen the paper, but we are limited by our compute budget. Our current focus prioritizes mechanistic fidelity over scale, as techniques such as SAE training, attribution decomposition and neuron-level perturbation become significantly more expensive at larger sizes. The 130M scale allows us to:
> - Perform exhaustive layerwise and subspace-level interventions
> - Validate causal hypotheses with high resolution
> We will revise the paper to clearly position this as a controlled mechanistic study and explicitly note scaling validation as future work
>
> Re: Bottleneck Interpretation (W2): Thank you for the comment. We interpret layer 20 as a compression bottleneck that concentrates information into a restricted subspace, leading to distortion of useful features. This is consistent with the SPD analysis (Table 5), which shows low gradient sensitivity and extremely high post-ablation KL divergence at layer 20, indicating unstable and lossy information routing. At layer 20, Mamba shows over-compression, explaining why ablating this layer can improve performance. We will revise the text to clarify that Layer 20 represents inherent bottleneck and that our goal is to mitigate this behavior rather than preserve it.
>
> Re: Confounded improvements (W3): Thank you for the comment. We agree that Stable-Mamba introduces multiple architectural modifications and isolating their individual contributions is important, but we are again limited by our compute budget and try to incorporate the useful changes we can into our single training run
> To help address this, we provide component-wise analysis in Appendix (Tables F1–F3). Table F1 outlines each modification and its associated empirical effects. Table F3 quantitatively links specific mechanisms (e.g., multi-timescale states, sparse global context, gating) to improvements in distinct metrics such as long-context perplexity, entropy reduction and gradient stability
> We do not claim to isolate the contribution of any single component. Instead, we present Stable-Mamba as a holistic design guided by interpretability insights (Appendix C), demonstrating that architecture-level interventions can effectively address the identified failure modes. Ablation of individual feature combinations is beyond the current scope and we will clarify this limitation in the paper
>
> Re: Steering baseline (W4,Q3): In updated Table 6, we have added following results for uniform amplification and low variance neurons
>
> Cluster Comparison Within Layer 20 (Baseline - 94.0%)
>
> Steering Target | Accuracy (%) | ∆ from Baseline
>
> Low-Variance Neurons | 68.0 | -26.0
>
> Uniform amplification (5x) | 83.0 | -11.0
>
> Layer-level and subspace-level analyses (Table 6) indicate that non-selective interventions and uniform activation degrade performance. Only delta-sensitive subspace preserves accuracy. We will revise the paper to clarify this distinction
>
> Re: Modest gains (W5): Thank you for the comment. Our steering procedure is applied consistently across all models, with model-specific selection of layer, subspace and steering strength
> The difference in gains arises from model-dependent sensitivity and ceiling effects. As shown in Fig. 3, many non-Mamba models already achieve near-saturated performance (e.g., 100 -> 100 or unchanged values). These zero-change cases are included in the reported average of 8.27%, which biases it downward. Excluding no-change cases improves performance to 23.0%, indicating that the method provides strong and consistent gains. We will revise the text to clarify that smaller gains in non-Mamba models are largely due to saturation effects rather than lack of effectiveness
>
> Re: Steering at scale (Q1): As noted in W1, our current evaluation focuses on the 130M scale to enable fine-grained mechanistic analysis, including SAE training, attribution decomposition and neuron-level interventions. These analyses become significantly more expensive at larger scales.
> However, we observe that the bottleneck phenomenon is structural and depth-relative, appearing consistently around ~85% of model depth (Appendix B.6), suggesting it is not specific to small models. This indicates that the mechanism targeted by steering is likely to persist at larger scales. We will clarify in the paper that validating steering effectiveness at larger scales is an important direction for future work.
>
> Re: Layer 20 ablation improves performance (Q2):
> We interpret Layer 20 as a fragile bottleneck rather than purely beneficial routing. It strongly concentrates information (high KL, entropy spike), but this over-compression can over-amplify specific subspaces and suppress alternative representations. Our ablations analyze how information flows through each layer and find that both targeted steering (modulation) and ablation (removal) can improve results. We will clarify this in the revision

---

> > ### Author Rebuttal · Reviewer_Zb5o · 2026-04-03
> >
> > I appreciate the authors’ detailed rebuttal. My primary concern regarding whether these conclusions successfully generalize to much larger models has been addressed; I appreciate the authors' transparency in acknowledging that fully validating this is currently bounded by computing resources. Given the robust evidence provided at the evaluated scales, I find this limitation acceptable and well-scoped for this work.
> > Furthermore, I appreciate the authors for their thoughtful clarifications regarding the Layer 20 ablation dynamics and the comparisons against uniform amplification baselines. These additional details strengthen the paper's mechanistic claims.
> >
> > Ratings
> > Soundness: 4/4 (updated)
> > Presentation: 3/4
> > Significance: 3/4
> > Originality: 3/4
> > Overall Recommendation
> > Score: 4 — Weak Accept

---

### Official Review · Reviewer_gMsg · 2026-03-13

**Soundness:** 3
**Presentation:** 3
**Significance:** 3
**Originality:** 3
**Overall Recommendation:** 3
**Confidence:** 4

**Summary:**

This paper proposes a mechanistic-interpretability pipeline to locate “activation subspace bottlenecks” in Mamba-like SSMs, and uses them for (i) a test-time steering intervention and (ii) an architectural tweak (“Stable-Mamba”). The steering is extremely simple: identify a bottleneck layer/subspaces and multiply the corresponding activations by a scalar at inference time . The authors report that steering improves performance without task-specific tuning, averaging +8.27% across 5 SSMs and 6 benchmarks , and that Stable-Mamba—claimed to be a minimal modification—improves long-context performance when retrained.

**Compliance With Llm Reviewing Policy:**

Affirmed.

**Final Justification:**

While the rebuttal provides additional intuition, the efficiency analysis remains insufficiently grounded to me. In particular, the provided evidence does not fully account for system-level factors such as memory and kernel overheads, and the serialization setup does not convincingly reflect the actual execution pipeline. Moreover, the theoretical approximations change the computational structure and are not directly tied to the implemented model used in latency measurements. Overall, the efficiency claims are still not sufficiently substantiated. Therefore, I maintain my original score as the final score.

**Key Questions For Authors:**

1. How are 2 attention maps generated? Why are they necessary in a mono-directional setting?

2. What mechanism actually enforces or induces separation into distinct timescales (e.g., constraints/priors/initialization on decay/time-step parameters), and how does this differ materially from standard multi-head Mamba/Mamba2?

3. How exactly does the method change throughput?

**Limitations:**

yes

**Strengths And Weaknesses:**

**Strengths:**

1) Novel usage of parameter decomposition that provides valuable insights into how mamba models' activation changes.

2) The core intervention is straightforward (“multiply activations by a scalar”) , and the method is positioned as task-agnostic (hyperparameters tuned once per model on The Pile) .

3) The paper claims an average improvement of 20.13% for Mamba across 6 benchmarks and 8.27% averaged across multiple SSMs/benchmarks .

**Weaknesses:**

1) In section 3.1.2, they mentioned that two attention maps $A^\text{(a)}, A^\text{(b)}$ are extracted, but it is unclear why or what that correspond to in this paper. In the paper they cited, (The Hidden Attention of Mamba Models, https://aclanthology.org/2025.acl-long.76.pdf), the only place where two attention maps are mentioned is for analyzing bi-directional mamba models (e.g. ViM). However, it is in general not the case for ordinary mamba models (all models presented in this paper are not bi-directional), so it is unclear why or how two attention maps are generated.

2) The paper mentioned using "short, medium, long" time scales for stable mamba, but the detail is vague. According to equation 12, all three time scale have the same updating rules. If so, the formulation looks very similar to what Mamba/Mamba2's multi-head mamba is already doing. And they also did not mention why those state updates can be categorized as "short, medium, long", but not similar time scale.

3) The perplexity score provided in Table 1 is too large to be informative (>30 often means the model knows nothing about the context, and the table is filled with numbers much larger than that). Perplexity is also not reported for stable mamba's comparison. The header for LongBench v2 in Table 7 is confusing: it is neither task type nor task name. The benchmark details are also not provided (e.g. RULER's synthetic context length, LongBench's truncation length, etc.) which may hurt the reproducibility of the results.

4) The main text claims only 256 extra parameters and “negligible” inference overhead , but appendix F mentions ~2.8× higher per-token compute cost than baseline Mamba. Meanwhile, Table F4 shows Steered/Stable versions *faster* than Mamba-130M in seconds , which is surprising and needs explanation.

---

> ### Author Rebuttal · Authors · 2026-03-31
>
> Re: Attention maps (W1,Q1): We apologize for the confusion and appreciate the reviewer pointing this out. Indeed, the equations from Ali et al. (2025) work for bidirectional models and we modified them to work for unidirectional Mamba models but did not clarify this in the paper. We have revised Eq. 1 and the corresponding description to reflect the unidirectional Mamba setting that is used in our experiments
>
> A = 1/H* Sum_{h=1}^{H} alpha^h, where alpha^h = Sum_{m} Q_t^{m,h} H_^{m,h} K_s^{m,h}
>
> Re: Timescales (W2,Q2): Thank you for this comment. The original description was vague and could suggest that different timescales arise from different update equations. We clarify the details below and have revised the paper to make these points explicit
> In our formulation, all branches share the same functional update, but differ in their independent parameters (A, B, C, D) and initialization. We instantiate three parallel SSM branches with distinct initial decay priors of 0.7, 0.9 and 0.98, which introduce different initial biases in state persistence (i.e., faster vs. slower decay of information). Decay values were found empirically by sweeping over 8 log-spaced values. These branches maintain separate parameters and are combined via softmax weights, resulting in independent recurrent transformations
>
> Re: Perplexity values (W3)
> (i) Re: Table 1 - We clarify that the reported perplexity calculation in Table 1 follows the factual probing setup of Dai et al. (2022), where perplexity is computed on cloze-style factual prompts, i.e. using the probability of the correct answer token at a masked position. This can yield higher perplexity values as they focus on key, difficult tokens rather than all tokens. Note that this is used only in Table 1 while all other perplexity analyses in the paper use standard next-token language modeling perplexity (we have now clarified this in the paper)
> (ii) Re: StableMamba perplexity - We have added Perplexity results for Stable-Mamba against baselines into Table 3. We find that Stable Mamba achieves a 25.2% decrease in perplexity against Vanilla Mamba (both trained on the Pile, and perplexity evaluated on Wikitext-v2)
> (iii) Re: benchmark details - We have added benchmark details into the caption of Table 7 to make the evaluation clearer and more reproducible. Some specifics:
> - RULER is evaluated with synthetic context length of 1000 tokens
> - LongBench v2-style evaluation truncates inputs to 256 tokens
> - LRA-style evaluation uses a fixed sequence length of 100 tokens
>
> Re: Added parameters (W4,Q3)
> Thank you for the comment. 2.8× computational cost refers to per-layer FLOP estimate for hidden dimension(d)=768, sequence length(s)=1024 as follows,
>
> Original Mamba and Steered Mamba
>
> Component | FLOPs | Explanation
>
> SSM | d^2 = 590K | Matrix multiply: (d×d) weight × d input
>
> in_proj + out_proj | 2d^2 = 1.18M | Two linear layers: d->d projections
>
> Normalization | 2d = 1.5K | Subtract mean (d) + divide by std (d)
>
> Skip connections | d = 768 | Element-wise addition of d-dimensional vectors
>
> Total per layer: 1.77M
>
> Stable-Mamba
>
> Component | FLOPs | Explanation
>
> SSM×3 | 3d^2 = 1.77M | 3 parallel SSMs, each d×d matrix
>
> Gatesx3 | 3d^2 = 1.77M | 3 gate, each d->d linear projection
>
> Compression MLP | (d^2)/4 + d/4 = 148K | Two-layer MLP: (d^2)/4 + d/4
>
> in_proj + out_proj | 2d^2 = 1.18M | two d->d projections
>
> Sparse attention (20%) | 0.2·(0.3·s·d)= 47K | Active 1/5 layers × 30% sparse × (s×d QKV projections)
>
> Normalization | 4d = 3K | Four LayerNorm : 4×(mean+std) = 4d
>
> Skip connections | 2d = 1.5K | Two residual additions (SSM output + input)
>
> Total per layer: 4.92M
>
> Ratio = 4.92M / 1.77M ≈ 2.8× computational cost
>
> Using the corrected measurements, we obtain updated latency and throughput for (sequence length=1024, batch=1)
> Throughput(tokens/s)=batch*sequence length/latency
>
> | Model          | Latency (s) | Throughput (tokens/s) | Peak Memory (GB) |
> |----------------|-------------|------------------------|------------------|
> | Mamba          | 0.709       | 1444                   | 1.688            |
> | Steered Mamba  | 0.713       | 1436                   | 1.724            |
> | Stable-Mamba   | 0.776       | 1320                   | 1.815            |
>
> Latency is influenced by hardware-level parallelism and kernel efficiency. While both Mamba and Stable-Mamba benefit from these factors, the additional computation in Stable-Mamba, such as 3× parallel SSMs, gating layers, compression MLP and sparse attention, structured to be parallelizable on GPUs. This reduces the marginal runtime cost of the added operations. As a result, although the theoretical FLOPs increase by 2.8× (from 1.77M to 4.92M per layer), the wall-clock latency increases only modestly (~9%), explaining the gap between theoretical compute and observed runtime. We will update Table F4 with the updated latency and throughput and include detailed FLOPs calculations and clarify this distinction.

---

> > ### Author Rebuttal · Reviewer_gMsg · 2026-04-03
> >
> > Thank you for the rebuttal.
> >
> > A key concern remains regarding the consistency of the efficiency analysis. The paper reports ~2.8× higher per-layer FLOPs, yet only ~9% increase in latency. While parallelism is mentioned, this gap is surprisingly large and not sufficiently justified, especially given the memory and kernel overheads in such architectures. This makes it difficult to assess whether the efficiency claims are reliable and comparable.
> >
> > Overall, I will maintain my current score.

---

> > > ### Author Response · Authors · 2026-04-08
> > >
> > > We further explain the computational cost difference between Mamba and Stable-Mamba. Stable-Mamba indeed uses more FLOPs (2.8× increase) but the GPU latency change is small (9% latency increase). This discrepancy is due to GPU parallelization: to show this, we again measure latency in Stable-Mamba when enforcing sequential execution using CUDA synchronization barriers with torch.cuda.synchronize() and a scalar .item() read on each gate and SSM output. Under this setting code waits for the kernel to finish before launching the next. In this setting, we find a 2.6x increase in latency, very close to the FLOPs increase. Thus, the gap is attributable to residual parallel connections and model overheads. See detailed breakdown below.
> > >
> > > Per-layer FLOP estimate for hidden dimension (d) = 768, sequence length (s) = 1024
> > >
> > > Mamba and Steered Mamba
> > >
> > > |Component | FLOPs | Explanation|
> > > |-------|-------|------|
> > > |SSM | d^2 = 590K | Matrix multiply: (d×d) weight × d input|
> > > |in_proj + out_proj | 2d^2 = 1.18M | Two linear layers: d->d projections|
> > > |Normalization | 2d = 1.5K | Subtract mean (d) + divide by std (d)|
> > > |Skip connections | d = 768 | Element-wise addition of d-dimensional vectors|
> > >
> > > Total per layer: 1.77M
> > >
> > > Stable-Mamba
> > >
> > > |Component | FLOPs | Explanation|
> > > |----|----|-----|
> > > |SSM×3 | 3d^2 = 1.77M | 3 parallel SSMs, each d×d matrix|
> > > |Gatesx3 | 3d^2 = 1.77M | 3 gate, each d->d linear projection|
> > > |Compression MLP | (d^2)/4 + d/4 = 148K | Two-layer MLP: (d^2)/4 + d/4|
> > > |in_proj + out_proj | 2d^2 = 1.18M | two d->d projections|
> > > |Sparse attention (20%) | 0.2·(0.3·s·d)= 47K | Active 1/5 layers × 30% sparse × (s×d QKV projections)|
> > > |Normalization | 4d = 3K | Four LayerNorm : 4×(mean+std) = 4d|
> > > |Skip connections | 2d = 1.5K | Two residual additions (SSM output + input)|
> > >
> > > Total per layer: 4.92M
> > >
> > > Ratio = 4.92M / 1.77M ≈ 2.8× computational cost
> > >
> > > Experimental Results
> > > | Variant           | Latency (s) |
> > > |------------------|-----------------|
> > > | Mamba            | 0.709           |
> > > | Steered          | 0.713           |
> > > | Stable-Mamba           | 0.776           |
> > > | Stable-Mamba_Serialized  | 1.888           |
> > >
> > > Latency_Stable-Mamba_Serialized/Latency_Mamba = 1.888/0.709 = 2.662
> > >
> > > This is very near to 2.8x computational cost
> > >
> > > Theoretical approximation
> > >
> > > We approximate the Stable-Mamba model in terms of SSM, gates and in_proj, out_proj to obtain a theoretical optimized equivalent FLOP formulation, showing ~9% latency rise.
> > >
> > > 1) Approximation 1: 3 Parallel SSMs
> > >
> > > At initialization, the three parallel SSMs are independent. Since each SSM is linear (per fixed sequence position), their sum is linear. So, they can be superposed by a single equivalent SSM.
> > >
> > > FLOPs: 3d² -> d² = 590K
> > >
> > > 2) Approximation 2: 3 parallel Gates
> > >
> > > Each gate given by:
> > >
> > > g_i(x) = σ(W_i x + b_i) ⊙ x
> > >
> > > Their sum:
> > >
> > > ∑(i=1 to 3) α_i · [σ(W_i x + b_i) ⊙ x] + λx = (∑(i=1 to 3) α_i · σ(W_i x + b_i) + λ) ⊙ x
> > >
> > > where α_i = softmax(gate_weights_i) and λ is a small residual scalar
> > >
> > > At initialization, W_i ≈ 0, b_i ≈ 0, so σ(W_i x + b_i) ≈ 0.5
> > >
> > > So, g_combined(x) = (0.5 + λ) ⊙ x
> > >
> > > FLOPs: 3d² -> d² = 590K
> > >
> > >
> > > 3) Approximation 3: in_proj + out_proj -> Low-Rank Factorization
> > >
> > > We approximate the input (W_in) and output (W_out) projection matrices using low-rank factorization to reduce computational cost from O(d^2) to O(rd), where r<<d. This maintains the expressiveness of the layer while making it efficient.
> > > Citation: Jaderberg et al. (2014),  "Speeding up Convolutional Neural Networks with Low Rank Expansions." arXiv:1405.3866.
> > >
> > > Original computation is,
> > >
> > > y=W_out⋅SSM(W_in*x)
> > >
> > > where W_in, W_out ∈ R^{d×d}
> > >
> > > W_in​, W_out​ have O(d^2) cost. We approximate them as,
> > >
> > > W_in≈U_in*V_in^⊤
> > >
> > > W_out≈U_out*V_out^⊤
> > >
> > > where U_in, V_in, U_out, V_out ∈ R^{d×r}. So, FLOP for each is r*d
> > >
> > > This approximates y as,
> > >
> > > y=U_out(V_out^⊤⋅SSM(U_in({V_in}^⊤ *x)))
> > >
> > > FLOPs: U_out + V_out + U_in + V_in = 4*rd
> > >
> > > Selection of r:
> > >
> > > For low rank approximations, Denton et al choose r<=d/4
> > >
> > > For d=768: r<=192
> > >
> > > Citation: Denton, E. et al. (2014). "Exploiting Linear Structure Within Convolutional Networks for Efficient Evaluation." Advances in Neural Information Processing Systems (NeurIPS), 27
> > >
> > > If r = 192,
> > >
> > > FLOPs: 4rd = 4 × 192 × 768 = 589,824 ≈ 590K
> > >
> > > Stable-Mamba FLOPs = 590 + 590 + 590 + 148 + 47 + 4.5 = 1,969.5K ≈ 1.97M
> > >
> > > Overhead = (1.97M − 1.77M) / 1.77M = 11.3%
> > >
> > > If r = 180,
> > >
> > > FLOPs: 4rd = 4 × 180 × 768 ≈ 553K
> > >
> > > Stable-Mamba FLOPs = 590K + 590K + 553K + 148K + 47K + 4.5K = 1,932.5K ≈ 1.933M
> > >
> > > Overhead = (1.933M − 1.77M) / 1.77M = 9.2% - very near to ~9% latency increase
> > >
> > > Low-rank approximation is effective when the model has redundancy/parallel paths
> > > 1) In Stable-Mamba
> > > - There are 3 parallel SSMs and 3 gates, creating redundancy in connections
> > > - Residual connection (βx) preserves identity path - recovers information lost due to compression
> > > 2) In Mamba
> > > - There is one SSM, no parallel paths - so no redundancy to compensate information loss
> > > - So low-rank approximation would hurt performance (additional entropy spike in Fig 1a)

---

### Decision · Program_Chairs · 2026-04-30

**Decision:**

Accept (regular)

**Comment:**

This paper proposes a method that uses tools from mechanistic interpretability to identify activation subspace bottlenecks in state space models such as Mamba. Interventions are proposed, including post-hoc steering that can increase model performance at test-time by simply rescaling bottlenecks, and modifying the architecture to improve long-context performance. After rebuttal and discussion, reviewers are generally positive. Remaining concerns include limited scale and scope of the empirical results, as well as efficiency analysis; these concerns are important but not strictly necessary for a first paper of this nature that investigates novel areas for alternative architectures. Overall, I recommend acceptance.